# STATISTICALLY MEANINGFUL APPROXIMATION: A THEORETICAL ANALYSIS FOR APPROXIMATING TURING MACHINES WITH TRANSFORMERS

## ABSTRACT

A common lens to theoretically study neural net architectures is to analyze the functions they can approximate. However, constructions from approximation theory may be unrealistic and therefore less meaningful. For example, a common unrealistic trick is to encode target function values using infinite precision. To address these issues, this work proposes a formal definition of statistically meaningful (SM) approximation which requires the approximating network to exhibit good statistical learnability. We study SM approximation for two function classes: boolean circuits and Turing machines. We show that overparameterized feedforward neural nets can SM approximate boolean circuits with sample complexity depending only polynomially on the circuit size, not the size of the network. In addition, we show that transformers can SM approximate Turing machines with computation time bounded by $T$ with sample complexity polynomial in the alphabet size, state space size, and $\log(T)$. We also introduce new tools for analyzing generalization which provide much tighter sample complexities than the typical VC-dimension or norm-based bounds, which may be of independent interest.

## 1    INTRODUCTION

Dating back to the seminal works on universal approximation (Cybenko, 1989; Hornik et al., 1989; Park & Sandberg, 1991; Leshno et al., 1993), a common way to theoretically study neural nets has been through their expressivity, which measures the ability of neural nets to approximate well-behaved functions. This perspective has shaped how researchers perceive different types of deep learning architectures: a basic way to theoretically justify new architectures is to study their approximation capabilities. This has led to a number of analyses studying universal approximation capabilities for various widely-used architectures, such as recurrent neural nets (RNNs) (Schäfer & Zimmermann, 2007), graph neural nets (Scarselli et al., 2008), convolutional networks (Bao et al., 2014; Zhou, 2020; Yarotsky, 2021), residual networks (Lin & Jegelka, 2018), transformers (Yun et al., 2019), and neural ODEs (Teshima et al., 2020; Zhang et al., 2020).

However, approximation theoretic results often misalign with more meaningful end-to-end guarantees, because models constructed in the literature often exhibit unrealistic properties. For example, a common technique in the universal approximation literature is to rely strongly on infinite-precision weights and activations, or exponentially many parameters to encode the desired function values (Hornik et al., 1989; Cybenko, 1989; Leshno et al., 1993; Lin & Jegelka, 2018; Yun et al., 2019; Sannai et al., 2019). This issue even arises outside of universal approximation, e.g., various papers demonstrate the ability of RNNs and transformers to simulate various computational models such as Turing machines and automata, but require strong reliance on arbitrary precision (Siegelmann & Sontag, 1995; Pérez et al., 2019; Korsky & Berwick, 2019; Bhattamishra et al., 2020). Infinite precision can inflate the expressivity of an architecture in a unrealistic and misleading way: for example, finite width RNNs with infinite precision can simulate Turing machines, but finite-precision, finite-width RNNs cannot, as implied by streaming lower bounds (Alon et al., 1999). As another example, Park et al. (2020) exploit infinite precision in the parameters to show that a neural net with parameter count sublinear in $n$ can memorize $n$ arbitrary input-label pairs. However, a simple counting argument reveals that this result cannot be proven using finite precision networks – there are $2^n$ input-labeling pairs, but only $2^{o(n)}$ finite precision networks with $o(n)$ parameters.

More broadly, the ideal theoretical perspective should consider not only whether target functions can be expressed, but also whether the constructed networks are plausibly *learnable*. Learnability is important because empirical settings do not operate in the infinite data, unbounded computation regime – they require fitting the target function with access to limited number of samples from an empirical distribution. The question of studying learnability can be decomposed into studying optimization and generalization.

Unfortunately, a rigorous analysis of optimization is unresolved even for simple two-layer nets (Mei et al., 2018). Generalization is more tractable, so we propose to study expressivity and generalization together.

Towards the goal of studying more meaningful notions of approximation, this work proposes the notion of *statistically meaningful (SM) approximation*. This definition requires not only the existence of an approximating network, but also that it has good statistical learnability. Consider a setting where the aim is to fit the target function $G$ using the approximating family $\mathcal{F}$ and a finite sample of training data. SM approximation requires existence of a loss whose empirical risk minimizer in $\mathcal{F}$ leads to a model with low approximation error in fitting $G$. We define the sample complexity of the approximation as the number of training samples needed to guarantee at most $\epsilon$ approximation error and study SM approximation with low sample complexity bounds. SM approximation essentially eliminates all statistical concerns for learnability (optimization-related concerns can remain).

We present two case studies on SM approximation. First, we demonstrate that overparameterized feed-forward neural nets can SM approximate boolean circuits with a low sample complexity that depends only on the intrinsic circuit size. Though it is simple to construct neural nets to approximate boolean circuits, bounding the sample complexity of the approximation is challenging. For example, standard norm-based generalization bounds for the naive construction scale exponentially in depth (Bartlett et al., 2017). Furthermore, VC dimension-based bounds would scale polynomially in the number of parameters in the network (Harvey et al., 2017), which is problematic because for practical optimization concerns, neural nets are typically overparameterized in terms of width (Zhang et al., 2016). In contrast, our sample complexity bound for SM approximation depends only on the intrinsic circuit size, up to logarithmic factors.

Our second case study is on SM approximating Turing machines with transformers. We consider a class of Turing machines with bounded computation time $T$ and construct encoder-decoder-based transformers (Vaswani et al., 2017) which SM approximate these Turing machines. The sample complexity of the approximation depends on a polynomial in $\log T$ and the sizes of the state space and the alphabet of the Turing machine. Though constructions for approximating Turing machines from prior work (Siegelmann & Sontag, 1995; Pérez et al., 2019; Bhattamishra et al., 2020) have not been formally studied from a sample complexity perspective, existing bounds would depend at least linearly on $T$. Furthermore, our construction only uses $\log\log T$ precision, compared to at least $\log T$ in prior works, allowing us to achieve the exponential improvement in the sample complexity.

Proving sample complexity guarantees for our statistically meaningful approximation results is nontrivial and requires additional insights, for both the constructions and the generalization analyses. To obtain our sample complexity bounds, we leverage a recent approach to bound generalization in terms of data-dependent notions of Lipschitzness (Wei & Ma, 2019b). We develop theoretical tools to convert a broad class of neural nets, with possibly large Lipschitzness, into ones with small Lipschitzness on the training data, by introducing a number of new layers that is linear in depth. Our result applies to neural nets where each entry in the hidden representations on the training data takes values from a finite set (e.g., binary entries), and may be of independent interest.

In summary, our contributions are: 1) we propose a new notion of statistically meaningful approximation, intended to provide more meaningful approximation guarantees by requiring that the approximating family have good statistical learnability; 2) we prove that feedforward neural nets can meaningfully approximate boolean circuits with sample complexity that depends polynomially on the width and depth of the circuit; and 3) we show that transformers can meaningfully approximate Turing machines with sample complexity logarithmic in the computation time.

## 1.1 RELATED WORKS

Classical approximation theory for neural networks has a long history. Hornik et al. (1989); Cybenko (1989), and Leshno et al. (1993) show that neural nets with one hidden layer are universal approximators but require the hidden layer size to grow exponentially in input dimension. Barron (1993) uses the Fourier transform to write target functions as infinite-width networks and subsamples neurons to obtain widths which depend only on target function properties. Lee et al. (2017); Ji et al. (2020) prove recent related developments in this direction of universal approximation.

Many works study benefits of deep networks over shallow ones (Bengio & Delalleau, 2011; Arora et al., 2016; Telgarsky, 2016; Eldan & Shamir, 2016; Daniely, 2017; Chatziafratis et al., 2020; 2019). Bengio & Delalleau (2011) show separation for exact representation, whereas Telgarsky (2016) shows separation for approximate representations with univariate inputs. Eldan & Shamir (2016) demonstrate high-dimensional functions that can be approximated by two-layer polynomial-sized neural networks, but cannot be approximated by one-layer neural nets with subexponential hidden units. Via reduction to certain complexity theo-

retic questions, Vardi & Shamir (2020) show that proving constant depth separations may be hard. Malach et al. (2021) analyze the relationship between optimization and approximability, showing in various settings that deeper networks cannot be optimized if shallow networks cannot approximate them. This demonstrates that depth separation results (Telgarsky, 2016) from approximation theory can be misleading in the sense that gradient descent anyways cannot optimize the deep networks used to construct the approximation.

Another area of study is on the ability of deep networks to memorize training data (Zhang et al., 2016; Yun et al., 2018; Park et al., 2020; Vershynin, 2020). Yun et al. (2018) show that $\Theta(n)$ parameters are sufficient to memorize $\Theta(n)$ training points for ReLU nets with at least 3 layers, and Park et al. (2020) reduce the parameter requirement to sublinear in $n$. Similar results have been proven for residual architectures (Hardt & Ma, 2016) and convolutional nets (Nguyen & Hein, 2018). Bartlett et al. (2019) analyze the VC-dimension of neural nets, leading to upper and lower bounds on the parameter count needed to fit training data. Other works study expressivity via connections to tensor approximation and sum-product networks (Cohen & Shashua, 2016; Cohen et al., 2016).

There is a long line of work on studying the ability of neural nets to recognize and represent formal languages. The seminal work of Siegelmann & Sontag (1995) shows that RNNs are Turing complete but leverages infinite precision in the hidden activations. Chen et al. (2018) extend this result to ReLU activations and study implications in language modeling. Many variants of transformers are shown to be Turing-complete, but these constructions also rely on arbitrary precision (Pérez et al., 2019; Bhattamishra et al., 2020). A number of recent works have also proven results for generating or recognizing formal languages with finite-precision neural nets (Weiss et al., 2018; Korsky & Berwick, 2019; Hewitt et al., 2020), but these results do not consider Turing machines or analyze statistical properties of their constructions. Bounding the sample complexity of SM approximation requires additional complications in both the construction and statistical analysis.

## 1.2 NOTATION

Let $f \circ g$ denote the composition of functions $f$ and $g$. For a family of functions $\mathcal{G}$, let $f \circ \mathcal{G} \triangleq \{f \circ g : g \in \mathcal{G}\}$ denote the family of compositions between $f$ and functions in $\mathcal{G}$. For a set $\mathcal{S}$ and function $f : \mathcal{S} \to \mathcal{Y}$, let $f(\mathcal{S})$ denote the set $\{f(s) : s \in \mathcal{S}\} \subseteq \mathcal{Y}$. We use $\mathbf{1}_d$ to denote the all-one's vector in $d$ dimensions, with the subscripted omitted if clear. For $i \in [d]$, we let $\mathbf{1}_d(i)$ denote the one-hot embedding in $d$-dimensions, which is 1 at index $i$ and 0 everywhere else. We use the notation $\widetilde{O}(\cdot)$ to hide poly-logarithmic factors in the argument. The notation $\lesssim, \gtrsim$ indicates the existence of a constant factor such that the inequality holds. $\asymp$ denotes that the $\gtrsim$ and $\lesssim$ relations simultaneously hold. We use poly$(\cdot)$ to indicate the existence of a polynomial in the argument which makes the equation true. For a set $\mathcal{A}$ (e.g., the set of alphabet symbols for a Turing machine) let $\mathcal{A}^*$ denote the set of all sequences of elements of $\mathcal{A}$, where sequence length can vary.

Let $P$ denote a distribution over a space of inputs $\mathcal{X}$. Let $\xi_1, ..., \xi_n$ be $n$ i.i.d. Rademacher variables sampled from $\{-1, +1\}$. The expected $n$-sample Rademacher complexity of $\mathcal{F}$ on $P$ is as follows: $\text{Rad}_{n,P}(\mathcal{F}) \triangleq \mathbb{E}_{(x_i)_{i=1}^n \overset{i.i.d}{\sim} P}\left[\mathbb{E}_{\xi_1, ..., \xi_n}\left[\sup_{F \in \mathcal{F}} \frac{1}{n}\sum_{i=1}^n \xi_i F(x_i)\right]\right]$, where $(x_i)_{i=1}^n$ denotes $n$ i.i.d. samples from $P$.

## 2 STATISTICALLY MEANINGFUL APPROXIMATION

We consider settings where we wish to approximate every member $G$ in a real-valued function class $\mathcal{G}$ with some function $F$ in function class $\mathcal{F}$. In this work, $\mathcal{F}$ is some family of neural networks. Fix a loss $\ell : \mathbb{R} \times \mathbb{R} \to [0,1]$. The classical definition of $\epsilon$-approximation states that $\mathcal{F}$ $\epsilon$-approximates $\mathcal{G}$ with respect to $\ell, P$ if for all $G \in \mathcal{G}$, there exists $F \in \mathcal{F}$ such that $\mathbb{E}_{x \sim P}[\ell(F(x), G(x))] \leqslant \epsilon$.

The issue with this classical notion of approximation is that in machine learning settings, we only have access to $G$, the function we wish to learn, through its values on a finite training set: $(x_i, G(x_i))_{i=1}^n$. If we disregard this fact, we could end up constructing functions $F$ which approximate $G$, but could have a number of unrealistic characteristics such as infinite precision. These drawbacks would mean that $F$ cannnot be realistically learned from the training sample.

This work studies a stronger notion of approximation, statistically meaningful (SM) approximation, to eliminate statistical concerns related to fitting $G$ on a finite sample. SM approximation, defined below, requires that $\mathcal{G}$ is learnable via empirical risk minimization using models from $\mathcal{F}$, when data is generated from $P$.

**Definition 2.1** (SM approximation). *$\mathcal{F}$ $\epsilon$-SM approximates $\mathcal{G}$ with respect to $\ell, P$ with sample complexity $n$ if there exists a loss $\bar{\ell} : \mathcal{F} \times \mathcal{X} \times \mathbb{R} \to [0,1]$ such that the following holds for all $G \in \mathcal{G}$:*

*Define $\widehat{F} \in \mathcal{F}$ to be the empirical minimizer of the loss $\bar{\ell}$ for fitting $G$ on an i.i.d. sample $(x_i, G(x_i))_{i=1}^n$ of $n$ examples labeled by $G$: $\widehat{F} \triangleq \operatorname{argmin}_{F \in \mathcal{F}} \frac{1}{n} \sum_{i=1}^n \bar{\ell}(F, x_i, G(x_i))$. Then with probability $0.99$ over the draw of $(x_i)_{i=1}^n$, $\widehat{F}$ approximates $G$ in the classical sense: $\mathbb{E}_{x \sim P}[\ell(\widehat{F}(x), G(x))] \leqslant \epsilon$.*

Definition 2.1 eases the statistical concerns associated with classical approximation theory: given a finite sample $(x_i, G(x_i))_{i=1}^n$, the empirical risk minimizer of $\bar{\ell}$ over $\mathcal{F}$ is guaranteed to $\epsilon$-approximate $G$ on the population distribution. It is important that the losses $\bar{\ell}$ (which can be interpreted as a training surrogate loss) and $\ell$ can be different, as this allows the empirical risk to include regularization.

Though Definition 2.1 may be reminiscent of PAC-learnability, there is a major conceptual difference: SM approximation unifies expressivity and generalization, whereas PAC-learnability is only concerned with generalization. The main focus of PAC-learnability is achieving a low loss relative to the best function in the hypothesis class, which is *assumed* to have 0 loss in the realizable case. SM approximation also requires *proving* that the best function in $\mathcal{F}$ achieves near-zero loss.

## 2.1 BACKGROUND AND TOOLS

To prove SM-approximation guarantees, Definition 2.1 requires a loss surrogate $\bar{\ell}$ such that the empirical risk minimizer of $\bar{\ell}$ on the training data can approximate functions in $\mathcal{G}$. The following proposition provides several conditions on $\bar{\ell}$ which lead to SM approximation guarantees.

**Proposition 2.2.** *For loss function $\ell : \mathbb{R} \times \mathbb{R} \to [0,1]$ and distribution $P$, suppose there exists a loss $\bar{\ell} : \mathcal{F} \times \mathcal{X} \times \mathbb{R} \to [0,1]$, intended as a surrogate loss for $\ell$, satisfying the following properties:*

*1) For all $F \in \mathcal{F}$, $x \in \mathcal{X}$, $y \in \mathbb{R}$, $\bar{\ell}(F, x, y) \geqslant \ell(F(x), y)$.*

*2) For all $G \in \mathcal{G}$, consider the function class $\mathcal{L}_G \triangleq \{x \mapsto \bar{\ell}(F, x, G(x)) : F \in \mathcal{F}\}$. Then the $n$-sample Rademacher complexity of $\mathcal{L}_G$ is bounded: $\operatorname{Rad}_{n,P}(\mathcal{L}_G) \leqslant \epsilon$.*

*3) For all $G \in \mathcal{G}$, there exists $F \in \mathcal{F}$ with small surrogate loss: $\mathbb{E}_{x \sim P}[\bar{\ell}(F, x, G(x))] \leqslant \epsilon$.*

*Then $\mathcal{F}$ $O\left(\epsilon + \frac{1}{\sqrt{n}}\right)$-SM approximates $\mathcal{G}$ with respect to $\ell, P$ with sample complexity $n$.*

By Proposition 2.2, it suffices that $\bar{\ell}$ upper bounds the target loss $\ell$ and has low complexity, and $\mathcal{F}$ approximates $\mathcal{G}$ with respect to $\bar{\ell}$, $P$ in the classical sense. The proof follows from standard techniques for bounding generalization based on Rademacher complexity and is provided in Section A.

**All-layer margin loss.** We introduce one particular construction for $\bar{\ell}$ used in subsequent sections, which is motivated by the all-layer margin generalization bound proposed by (Wei & Ma, 2019b). This bound is based on data-dependent Lipschitzness measures (Nagarajan & Kolter, 2019; Wei & Ma, 2019a), and can provide stronger guarantees than classical norm-based bounds (Neyshabur et al., 2015; Bartlett et al., 2017; Neyshabur et al., 2017; Golowich et al., 2018).

We focus on the binary classification setting, where $G(x) \in \{0,1\}$, and study approximation with respect to the 0-1 loss $\ell_{0\text{-}1}(z,y) \triangleq \mathbb{1}((y-0.5)z \leqslant 0)$ where $y \in \{0,1\}$ is assumed to be a binary label. We consider a family of functions $\mathcal{F}$ parameterized by $p$-dimensional parameters $\theta \in \Theta \subseteq \mathbb{R}^p$, with a general architecture function $F : \mathcal{X} \times \mathbb{R}^p \to \mathbb{R}$. Thus, $\mathcal{F} = \{x \mapsto F(x, \theta) : \theta \in \Theta\}$, and we sometimes use $\theta$ to identify an element of $\mathcal{F}$. Throughout the paper, we define $\Theta$ as a $\|\cdot\|_1$-norm bounded set: $\|\theta\|_1 \leqslant \alpha$, $\forall \theta \in \Theta$. We define the parameter-based all-layer margin $\rho_F : \mathbb{R}^p \times \mathcal{X} \times \{0,1\} \to \mathbb{R}$ as follows:

$$\rho_F(\theta, x, y) \triangleq \min \|\delta\|_2$$
$$\text{subject to } (y - 0.5) \cdot F(x, \theta + \delta) \leqslant 0 \tag{2.1}$$

We omit the architecture from the subscript when it is clear from context. This quantity measures the stability of the model around an input $x$ in parameter space. As is the case for the standard output margin, a larger all-layer margin, or better stability, implies better generalization.

We modified the definition in (Wei & Ma, 2019b) to consider perturbations $\delta$ in parameter space, whereas Wei & Ma (2019b) consider perturbations to the hidden layers. The parameter-space formulation is simpler and subsumes the results in (Wei & Ma, 2019b). Our formulation also accounts for weight sharing, which is important for our Turing machine results, whereas the formulation of (Wei & Ma, 2019b) could not.

A key and immediate property of the all-layer margin is that it is strictly positive if and only if $F(x, \theta)$ predicts the correct label. We can leverage this property to construct a surrogate loss. For some parameter

$\gamma$ intended to lower bound the all-layer margins, we define the loss $\bar{\ell}_\gamma$ as follows:

$$\bar{\ell}_\gamma(\theta,x,y) = \begin{cases} 1 \text{ if } \rho(\theta,x,y) \leqslant 0 \\ 1 - \frac{\rho(\theta,x,y)}{\gamma} \text{ if } 0 < \rho(\theta,x,y) \leqslant \gamma \\ 0 \text{ if } \rho(\theta,x,y) \geqslant \gamma \end{cases} \tag{2.2}$$

Note that $\bar{\ell}_\gamma$ composes the classical ramp loss, which is used to prove margin-based generalization complexity bounds, with the value of the all-layer margin. By our construction, it immediately follows that $\bar{\ell}_\gamma(\theta,x,G(x)) \geqslant \ell_{\text{0-1}}(F(x,\theta),G(x))$, as is required of a surrogate loss.

We show that to obtain sample complexity bounds for SM approximation of $\mathcal{G}$ in a classification setting, it suffices to prove that functions in $\mathcal{F}$ can fit labels of $G \in \mathcal{G}$ with large all-layer margin.

**Lemma 2.3.** *Fix any neural net architecture $F : \mathcal{X} \times \mathbb{R}^p \to \mathbb{R}$, and define $\mathcal{F}_\alpha \triangleq \{x \mapsto F(x,\theta) : \theta \in \Theta\}$, where we assume $\Theta \subseteq \mathbb{R}^p$ is such that $\|\theta\|_1 \leqslant \alpha$ for all $\theta \in \Theta$. Fix $\epsilon \geqslant 0$. Suppose that for all $G \in \mathcal{G}$, there exists $\theta \in \Theta$ such that the following holds:*

$$\mathbb{E}_{x \sim P}[\mathbb{1}(\rho(\theta,x,G(x)) < \gamma)] \leqslant \epsilon \tag{2.3}$$

*Then $\mathcal{F}_\alpha$ $\epsilon$-SM approximates $\mathcal{G}$ with respect to $\ell_{\text{0-1}},P$ with sample complexity $\widetilde{O}\left(\frac{1}{\epsilon^2}\left(\frac{\alpha^2 \log(p)}{\gamma^2} + 1\right)\right)$.*

We note that $\widetilde{O}$ hides poly-logarithmic factors in the arguments, in this case, polylog$(\frac{\alpha^2 \log(p)}{\gamma^2 \epsilon^2})$ factors. The proof closely follows (Wei & Ma, 2019b), is deferred to Section A. In Section A, we also state a generalization bound for 0-1 loss based on (2.1), which may be of independent interest. We use (2.2) and Lemma 2.3 to prove that neural nets can SM approximate Boolean circuits and Turing machines.

## 3 SM APPROXIMATION OF BOOLEAN CIRCUITS WITH FEEDFORWARD NETS

This section shows that feedforward neural nets can SM approximate Boolean circuits with sample complexity that depends polynomially on the size of the circuit. A boolean circuit $G : \{0,1\}^m \to \{0,1\}$ on $m$ inputs bits is described by a directed acyclic graph, with vertices of this graph referred to as "gates". The graph contains $m$ input gates of indegree 0, which are identified with the input bits. The remaining gates each compute a boolean function taking values at their parents as arguments, and a designated output gate produces the output of the entire circuit. We consider boolean circuits consisting of AND, OR, and NOT gates, which compute the corresponding boolean functions on 2, 2, and 1 inputs, respectively and are sufficient to compute any boolean function (Savage, 1998). We also allow identity (ID) gates, which take 1 input and output the same value.

We consider layered circuits, where we can partition the gates into layers such that the only edges in the graph occur from gates in layer $i$ to gates in layer $i+1$ for some $i$. Note that we can transform any boolean circuit into a layered one by adding ID gates. Letting $q$ denote the number of layers and $r$ the maximum number of gates in any layer, we say that the circuit has depth $q$ and width $r$. We say that a circuit with $s$ total gates has size $s$. Our convention will be that the set of input gates is considered a layer, so $r \geqslant m$. We consider the following class of boolean circuits:

$$\mathcal{G}_{q,r,s} = \{G : \{0,1\}^m \to \{0,1\} : G \text{ computed by circuit with depth } q, \text{ size } s, \text{ and width } r\}$$

We will approximate $\mathcal{G}_{q,r,s}$ using a family of width $w$, depth $d$ feedforward ReLU nets parameterized by linear weights and biases $\theta = (W_0, b_0, \dots, W_d, b_d)$ computed as follows: $F_{w,d}(x,\theta) = W_d\phi(W_{d-1}\phi(\cdots \phi(W_0 x + b_0)\cdots) + b_{d-1}) + b_d$, where all intermediate layers have width $w$ for simplicity and $\phi$ denotes the coordinate-wise ReLU activation. The weight parameters are set so that for $1 \leqslant i \leqslant d-1$, $W_i \in \mathbb{R}^{w \times w}$, $W_0 \in \mathbb{R}^{w \times m}$, and $W_d \in \mathbb{R}^{1 \times w}$. The bias parameters are such that $b_i \in \mathbb{R}^w$ for $0 \leqslant i \leqslant d-1$, and $b_d \in \mathbb{R}$. To control the sample complexity, we restrict our attention to the set of parameters with total $\|\cdot\|_1$-norm bounded by $\alpha$, giving the following function class:

$$\mathcal{F}_{w,d,\alpha} = \{x \mapsto F_{w,d}(x,\theta) : \|\theta\|_1 \leqslant \alpha\}$$

The following theorem states that feedforward neural nets can statistically meaningfully approximate boolean circuits with sample complexity polynomial in the circuit size.

**Theorem 3.1.** *Consider the class $\mathcal{G}_{q,r,s}$ of size-$s$, width-$r$, and depth-$q$ layered boolean circuits, and the class $\mathcal{F}_{w,d,\alpha}$ of neural nets above. Suppose $w \gtrsim r$, $\alpha \asymp s$, and $d \asymp q$.*

*Then for all $\epsilon > 0$ and any input distribution $P$ over $\{0,1\}^m$, $\mathcal{F}_{w,d,\alpha}$ $\epsilon$-SM approximates $\mathcal{G}$ with respect to $\ell_{\text{0-1}},P$ with sample complexity $\text{poly}(s)\widetilde{O}\left(\frac{\log(wd)}{\epsilon^2}\right)$.*

We note that the bound in Theorem 3.1 only scales logarithmically in the width $w$ of the network, even if $w$ is arbitrarily greater than the circuit width $r$. This ensures that even heavily overparameterized nets will have low sample complexity of the approximation.

For this setting, the all-layer margin loss in (2.2) is essential for proving tight sample complexity bounds, as other surrogate losses $\bar{\ell}$ would give weaker results. For example, if we choose $\ell_{0\text{-}1}$ as the surrogate loss, VC-dimension bounds (Harvey et al., 2017) imply that $\mathcal{F}_{w,d,\alpha}$ statistically meaningfully approximates $\mathcal{G}_{q,r,s}$ with sample complexity scaling in $\mathrm{poly}(wq)$ under the conditions of Theorem 3.1. This suffers a polynomial dependence on the overparameterized width $w$, which is not ideal for realistic settings, where neural nets are often wider than necessary to facilitate optimization. In contrast, our dependence on $w$ is logarithmic. Another possible surrogate loss is the *output* margin-based ramp loss, which can be used to prove norm-based sample complexities (Bartlett et al., 2017). However, these bounds depend on $\prod_{i=1}^{d} \|W_i\|_{\mathrm{op}}$ (or related quantities), which would be exponentially large in $d$ for the naive construction in Section 3.1.

### 3.1 PROOF SKETCH FOR THEOREM 3.1

There are two key steps in the proof. First, given any layered circuit $G \in \mathcal{G}$, we construct a neural net that directly simulates $G$ by computing the layers of $G$ one-by-one, which is simple to do by directly constructing ReLU and linear layers to simulate the AND, OR, NOT, and ID gates.

**Lemma 3.2.** *In the setting of Theorem 3.1, let $G$ denote the layered boolean circuit, which we aim to compute using a neural net. Let $g_i : \{0,1\}^{r_{i-1}} \to \{0,1\}^{r_i}$ denote function computed between the $i-1$-th and $i$-th layers of $G$, which we assume have $r_{i-1}$ and $r_i$ gates, respectively, so $G = g_{q-1} \circ \cdots \circ g_1$. Then there exist functions $f_1, ..., f_{q-1}$, where each $f_i$ is computed by a feedforward ReLU net with two linear and activation layers, such that for all $i \in [q-1]$ and $x \in \{0,1\}^m$*

$$f_i \circ \cdots \circ f_1(x) = g_i \circ \cdots \circ g_1(x)$$

*Thus, the composition $F(\cdot, \theta) \triangleq f_{q-1} \circ \cdots \circ f_1$ satisfies $F(x, \theta) = G(x)$ for all $x \in \{0,1\}^m$. Note that we omitted the dependency of $f_{q-1}, ..., f_1$ on parameters $\theta$ for simplicity.*

**Lower bounding all-layer margin.** The next step for proving SM approximation is to construct a loss $\bar{\ell}$ so that the empirical risk minimizer of $\bar{\ell}$ on the training data has good sample complexity. This crucially requires the all-layer margin tool developed in Section 2.1, as other complexity measures (e.g. norm-based) would not give good sample complexity bounds.

Recall that the all-layer margin $\rho_F(\theta, x, G(x))$ measures the stability of the output $F(x, \theta)$ to perturbations in to $\theta$, and, by Lemma 2.3, it suffices to show that $F$ has large all-layer margin on $x \in \{0,1\}^m$. Unfortunately, we cannot guarantee that the naive construction from Lemma 3.2 has large all-layer margin without further modifications. To remedy this issue, Theorem D.6 introduces a generic way to convert the model $F(\cdot, \theta)$, with possibly small all-layer margin on $x \in \{0,1\}^m$, into a new architecture and parameter set $F'(\cdot, \theta')$, with provably large all-layer margin on $x \in \{0,1\}^m$, such that $F'(x, \theta') = F(x, \theta)$ on all inputs $x \in \{0,1\}^m$. The construction relies on introducing new layers to $F$ to obtain $F'$ and increases the total number of layers by only a constant factor. This step of the proof is formally stated in the following lemma.

**Lemma 3.3.** *In the setting of Lemma 3.2, let $F(\cdot, \theta) = f_{q-1} \circ \cdots \circ f_1$ be the neural net with parameters $\theta$ constructed to compute the circuit $G$. There exist "correction functions" $\zeta_1, ..., \zeta_{q-2}$, where $\zeta_i$ is computed by a neural net with two activation and linear layers, such that the composition*

$$F'(\cdot, \theta') \triangleq f_{q-1} \circ \zeta_{q-2} \circ f_{q-2} \circ \cdots \circ \zeta_1 \circ f_1$$

*has large all-layer margin. Here $\theta'$ denotes the collection of all parameters. Concretely, $\rho_{F'}(\theta', x, G(x)) \geq \frac{1}{\mathrm{poly}(s)}$ for all $x \in \{0,1\}^m$. Note that we omitted the dependency of $f_i, \zeta_i$ on parameters $\theta'$ for simplicity.*

We convey the core intuitions for Lemma 3.3 in a simplified toy setting as follows. Consider the case where we start with an initial architecture $f$ computing $f(x, (W_1, ..., W_d)) = \left( \prod_{i=1}^{d} W_i \right) x - 0.5$, where $W_i \in \mathbb{R}$. In this simplified setting, we consider $W_i = 1 \ \forall i$. For input $x = 1$ and target $y = 1$, the all-layer margin is small: $\rho_f((1, ..., 1), 1, 1) \lesssim \frac{1}{\sqrt{d}}$, where the architecture is in the subscript. Indeed, choosing $\delta_i = \frac{3}{d}$, we have $f(1, (1 - \frac{3}{d}, ..., 1 - \frac{3}{d})) = (1 - \frac{3}{d})^d - 0.5 \approx \exp(-3) - 0.5 < 0$. Thus, by the definition of all-layer margin, $\rho_f((1, ..., 1), 1, 1) \leq \sqrt{\sum_i \delta_i^2} \lesssim \frac{1}{\sqrt{d}}$.

Now we will insert ReLU layers in $f$ to increase the all-layer margin to $\Omega(1)$. We use ReLU layers to implement the round function, which has the key property that $\mathrm{round}(z) = 1 \ \forall z \geq 2/3$.

**Proposition 3.4.** *For any* $z \in \mathbb{R}$*, we can implement the function* $\text{round}(z) = \begin{cases} 0 & \text{if } z < 1/3 \\ 3x-1 & \text{if } 1/3 \leqslant z < 2/3 \\ 1 & \text{if } z \geqslant 2/3 \end{cases}$

*via a feedforward ReLU net, as follows:* $\text{round}(z) = 3\phi(z-1/3) - 3\phi(z-2/3)$.

We consider the following function $\widetilde{f}$, which inserts round between every layer in $f$:

$$\widetilde{f}(x,(W_1,...,W_d)) = \text{round}(W_d \text{round}(W_{d-1}\cdots\text{round}(W_1 x)\cdots)) - 0.5 \tag{3.1}$$

For this demonstration, we ignore the parameters of round, though the actual proof considers these parameters. The following claim shows that (3.1) preserves the output of $f$ while increasing the all-layer margin:

**Claim 3.5.** *In the setting above, it holds that* $\widetilde{f}(1,(1,...,1)) = f(1,(1,...,1))$ *and* $\rho_{\widetilde{f}}((1,...,1),1,1) \geqslant \frac{1}{3}$.

This reflects a significant increase in the all-layer margin, while only increasing depth by a constant factor. The proof is simple: we observe that if $\delta_i \leqslant \frac{1}{3}$ for all $i$, the function output will not change because $\text{round}(z) = 1 \; \forall z \geqslant \frac{2}{3}$. This immediately gives the all-layer margin lower bound $\frac{1}{3}$.

To apply this construction more generally, we note that round corrects errors in previous layers. In the more general setting, we insert "correction functions" $\zeta$ between each layer satisfying the key property that $\zeta(h') = h$ if $h$ is the intended output of the layer and $h'$ is any perturbed value satisfying $\|h'-h\|_2 \leqslant \frac{1}{3}$. Since intended outputs of layers in the function constructed by Lemma 3.2 are binary-valued in $\{0,1\}^w$ because $F$ simulates a boolean circuit, we can simply apply the function round constructed in Proposition 3.4 elementwise as the correction function. By the construction, this can be implemented by adding two additional feedforward ReLU layers per correction function. Following the intuition for Claim 3.5, we prove that inserting these correction functions guarantees a large all-layer margin (Theorem D.6) on all $x \in \{0,1\}^m$. This leads to the proof of Lemma 3.3. We can complete the proof of Theorem 3.1 by invoking Lemma 2.3, as shown in Section B.

## 4 SM APPROXIMATION OF TURING MACHINES WITH TRANSFORMERS

In this section, we show that transformers SM approximate Turing machines with computation time bounded by $T$, using sample complexity polynomial in $\log T$ and the state space and alphabet sizes of the Turing machine. Constructions from prior work would require the sample complexity of the approximation to be linear in $T$ (Siegelmann & Sontag, 1995; Chen et al., 2018; Pérez et al., 2019; Bhattamishra et al., 2020). Thus, we obtain an exponential improvement in the dependency on $T$.

We briefly describe a Turing machine; see (Sipser, 2013) for a more thorough survey. A Turing machine is a model for computation specified by a tuple $(\mathcal{Z}, \mathcal{A}, S, \mathcal{Z}_{\text{term}})$ containing a set of states $\mathcal{Z}$, a tape alphabet $\mathcal{A}$, a transition function $S: \mathcal{Z} \times \mathcal{A} \to \mathcal{Z} \times \mathcal{A} \times \{-1,+1\}$, and set of terminal states $\mathcal{Z}_{\text{term}}$ indicating accept or reject. For simplicity, we assume the Turing machine has a single tape, as any single-tape Turing machine can simulate a multi-tape one with only quadratic increase in runtime (Sipser, 2013). Given an input $x \in \mathcal{A}^*$ recorded on the left-most part of the tape, the Turing machine performs computation in a sequence of timesteps. In each timestep, the machine determines the next state, symbol to write, and direction to move the head via the transition function.

We let $\text{TM}_{(\mathcal{Z}, \mathcal{A}, S, \mathcal{Z}_{\text{term}})}$ denote the function computed by the Turing machine, which produces an output in $\{0,1\}$ (if the machine halts). Fixing the alphabet $\mathcal{A}$, we consider the class of binary functions computed by Turing machines with at most $k$ states terminating in $T$ steps:

$$\mathcal{G}_{k,T} \triangleq \{x \mapsto \text{TM}_{(\mathcal{Z}, \mathcal{A}, S, \mathcal{Z}_{\text{term}})}(x) : |\mathcal{Z}| \leqslant k, \text{ and } \forall x \in \mathcal{X}, \text{TM}_{(\mathcal{Z}, \mathcal{A}, S, \mathcal{Z}_{\text{term}})} \text{ terminates in } T \text{ steps }\} \tag{4.1}$$

### 4.1 TRANSFORMER ARCHITECTURE FOR SM-APPROXIMATING TURING MACHINES

We study approximation of $\mathcal{G}$ with a family of architectures consisting of both an encoder and decoder component (Vaswani et al., 2017), described as follows. The encoder architecture is simple and only performs an embedding of the input symbols, using learnable symbol embeddings $E \in \mathbb{R}^{w \times |\mathcal{A}|}$ and fixed positional encodings $\beta(1), \beta(2), ... \in \mathbb{R}^w$. Given input $x \in \mathcal{A}^*$ with $m$ symbols, the encoder produces $m$ output vectors in $\mathbb{R}^w$ via $\text{Enc}_i(x,E) = E_{:,x_i} + \beta(i)$, where $\text{Enc}_i$ denotes the output of the encoder at the $i$-th position.

The decoder iteratively computes an output, running for $T$ steps. We define a transformer layer of the decoder as a sequence of modules consisting of decoder self-attention, followed by encoder-decoder attention, followed by three feedforward ReLU layers.

**Attention layers.** Attention layers consist of key, value, and query functions $K, V, Q$, each of which computes a linear transformation. We omit parameters here for simplicity. Restricted to a single decoder timestep, the attention layer takes two types of inputs: a sequence of previously-computed representations $h_1, ..., h_i$, and a current input representation $h'$. The layer first applies the key, value, and query functions as follows:

$$\tau_0, \tau_1, ..., \tau_i = Q(h')^\top K_0, Q(h')^\top K(h_1), ..., Q(h')^\top K(h_i)$$
$$v_0, v_1, ..., v_i = V_0, V(h_1), ..., V(h_i)$$

where $K_0$ and $V_0$ are fixed "null" key and value vectors which are learned parameters of the layer. Letting $\mathcal{J}$ denote the set of indices $\{j : \tau_j = \max\{\tau_0, ..., \tau_i\}\}$, the attention layer performs hard-max attention (Pérez et al., 2019) to compute the output, as follows:

$$\text{Attn}(h', (h_1, ..., h_i)) = h' + \frac{1}{|\mathcal{J}|} \sum_{j \in \mathcal{J}} v_j$$

Our theory also applies to the standard softmax attention used in practice, but we focus on the hard-max case for a simpler proof. Let $h_t^{(j)}$ denote the representation computed by the $j$-th layer of the decoder at timestep $t$. At timestep $i$, decoder self-attention at the $(j+1)$-th layer computes $\text{Attn}(h_i^{(j)}, (h_1^{(j)}, ..., h_i^{(j)}))$. Letting $e_1, ..., e_m$ denote the encoder outputs, encoder-decoder self-attention at the $(j+1)$-th layer and $i$-th step would compute $\text{Attn}(h_i^{(j)}, (e_1, ..., e_m))$.

**Transformer layers.** We use feedforward layers which apply 3 standard ReLU layers, as follows: $\text{FF}(h) = \phi(W_3 \phi(W_2 \phi(W_1 h + b_1) + b_2) + b_3)$. Our theory also allows for residual feedforward layers, and the architecture here is chosen mainly to simplify the construction.

A transformer layer applies these constructions in sequence. Letting $H_i^{(j)} = (h_1^{(j)}, ..., h_i^{(j)})$ denote the output after the $j$-th transformer layer for timesteps $1 \leq t \leq i$, and $\theta^{(j)}$ the parameters of the layer, we compute

$$h_i^{(j+1, \text{dec})} = \text{Attn}(h_i^{(j)}, H_i^{(j)}, \theta^{(j+1, \text{ dec-attn})})$$
$$h_i^{(j+1, \text{enc})} = \text{Attn}(h_i^{(j+1, \text{dec})}, (e_1, ..., e_m), \theta^{(j+1, \text{ enc-attn})})$$
$$\text{Tr}(h_i^{(j)}, H_i^{(j)}, (e_1, ..., e_m), \theta^{(j+1)}) = \text{FF}(h^{(j+1, \text{enc})}, \theta^{(j+1, \text{ ff})})$$

Note that we included the explicit dependence of the attention layers on the parameters for completeness. We now set $h_i^{(j+1)} = \text{Tr}(h_i^{(j)}, H_i^{(j)}, (e_1, ..., e_m), \theta^{(j+1)})$.

**Decoder outputs.** We consider $d$-layer decoders, so $o_i \triangleq h_i^{(d)}$ denotes the output of the decoder at time $i$, which is also inputted to the decoder at time $i+1$ as follows: $h_{i+1}^{(0)} = h_i^{(d)} + \beta(i+1)$. The initial decoder input $h_0^{(0)}$ is a trainable parameter. The decoder runs for a fixed number of timesteps $T'$ and produces the prediction $\theta_{\text{cls}}^\top h_{T'}^{(d)}$. For simplicity, we assume $T' = T$, the computation time of the Turing machine family.

Note that our architecture allows long (length $T$) decoding sequences, whereas typical architectures in practice use decoding sequences with roughly the same length as the input (Vaswani et al., 2017). The architecture we study is similar to ones studied by (Pérez et al., 2019; Bhattamishra et al., 2020).

We use $x \mapsto F_{w,d,T}(x, \theta)$ to denote the described transformer architecture with parameters $\theta$, $w$-dimensional hidden layers, $d$ transformer layers in the decoder, and $T$ decoder steps. This leads to the following class of transformer functions: $\mathcal{F}_{w,d,\alpha,T} = \{x \mapsto F_{w,d,T}(x, \theta) : \|\theta\|_1 \leq \alpha\}$. The following theorem states that this class of transformers SM approximates the Turing machine family $\mathcal{G}$ defined in (4.1) with sample complexity polynomial in $\log T$, $k$ and $|\mathcal{A}|$.

**Theorem 4.1.** *In the setting above, consider the class $\mathcal{G}$ of functions computed by Turing machines with at most $k$ states, alphabet $\mathcal{A}$, and computation time bounded by $T$ steps for inputs $x \in \mathcal{X}$. Suppose that $w \gtrsim k|\mathcal{A}| + \log T$, $d \asymp \log T$, and $\alpha = \text{poly}(k, |\mathcal{A}|, \log T)$.*

*Then for all $\epsilon > 0$ and any input distribution $P$ over $\mathcal{X}$, $\mathcal{F}_{w,d,\alpha,T}$ $\epsilon$-SM approximates $\mathcal{G}$ with respect to $\ell_{0\text{-}1}, P$ with sample complexity $\text{poly}(k, |\mathcal{A}|, \log T) \widetilde{O}\left(\frac{\log(wd)}{\epsilon^2}\right)$.*

As with Section 3, we set the surrogate loss $\bar{\ell}$ in Definition 2.1 to be the all-layer margin loss defined in Section 2.1. Commonly-used alternatives for the surrogate loss would not suffice for either our construction or ones in prior work (Siegelmann & Sontag, 1995; Chen et al., 2018; Pérez et al., 2019; Bhattamishra et al., 2020). First, the VC dimension of $\mathcal{F}_{w,d,\alpha,T}$ is at least $\Omega(wT)$. This is because transformer architectures

which contain a decoder component can express RNNs, which by lower bounds have VC dimension at least $wT$ (Koiran & Sontag, 1998). This indicates that using $\ell_{\text{0-1}}$ as the surrogate loss would lead to sample complexities that are suboptimal in both the overparameterized width $w$ and the computation $T$. Second, the correct norm-based Rademacher complexity bound to use for transformers is unclear; however, the RNN-based equivalent would scale with the $T$-th power of some parameter norm, or exponentially in $T$. Thus, as in Section 3, the all-layer margin surrogate loss (2.2) is essential for obtaining our sample complexity bounds.

### 4.2 Proof Sketch for Theorem 4.1

Following Lemma 2.3, our goal is to construct a transformer which can simulate Turing machines with large all-layer margin, namely, $\Omega\left(\frac{1}{\text{poly}(k,|\mathcal{A}|,\log T)}\right)$. The fundamental limitation of prior work (Pérez et al., 2019) towards attaining this is that the positional embeddings are required to store values as small as $\frac{1}{\text{poly}(T)}$. Our construction cannot afford to rely on values this small – informally, if the construction relies on the exact values of these small entries, then the all layer margin would be at most $\frac{1}{\text{poly}(T)}$ because perturbing the layer by the small entries could change the prediction. Instead, we propose using $\text{Bin}(i)$, the binary encoding of $i$ in $\lceil \log T \rceil$ bits, as the positional encoding for timestep $i$. This allows us to use unique positional encodings for each timestep which do not rely on arbitrary precision.

We describe the construction of the transformers. Fix a Turing machine $G \in \mathcal{G}$. We first require notation to describe the computation of $G$. For input $x \in \mathcal{X}$, we define $z_i(x)$, $a_i(x)$ to be the Turing machine state and symbol under the tape head at the conclusion of step $i$. We let $l_i(x)$ denote the location of the Turing machine head at the conclusion of step $i$. During the timestep, the Turing machine computes $S(z_{i-1}(x), a_{i-1}(x))$, writes a new symbol under the head at location $l_{i-1}(x)$, and moves the head either left or right. Let $u_i(x)$ denote the symbol written during timestep $i$, and $q_i(x) \in \{\text{left},\text{right}\}$ the movement direction of the head.

Following (Pérez et al., 2019) with several key modifications, we simulate the Turing machine using the transformer as follows. Each timestep will maintain the invariance that $o_i$ contains an encoding of $z_i(x), a_i(x)$, and $l_i(x)$. Given that this invariance holds until timestep $i$, the transformer simulates timestep $i+1$ of the Turing machine with the following steps:

1) Use feedforward layers to apply transition $S$ on $z_i(x)$ and $a_i(x)$, which can be read from $o_i$, to obtain $z_{i+1}(x)$, $u_{i+1}(x)$, and movement direction $q_{i+1}(x) \in \{\text{left, right}\}$.

2) Using feedforward layers, compute $l_{i+1}(x)$ from $q_{i+1}(x)$ and the encoding of $l_i(x)$ in $o_i$.

3) Compute $a_{i+1}(x)$. We use decoder self-attention to search over past timesteps which wrote to $l_{i+1}(x)$. Our aim is to find $u_{i'}(x)$, where $i' = \max\{j \leqslant i+1 : l_{j-1}(x) = l_{i+1}(x)\}$. We implement a binary search over past timesteps $j$, which is needed to find the *largest* $j \leqslant i+1$ where $l_{j-1}(x) = l_{i+1}(x)$. The binary search can be implemented with $O(\lceil \log T \rceil)$ decoder self-attention layers, and the construction ensures large all-layer margin.

4) If no such $i'$ from the previous timestep existed, we check whether $l_{i+1}(x)$ contained an input symbol using encoder-decoder attention and copy this input symbol if so.

5) If no symbols were found in 3) or 4), $l_{i+1}(x)$ must contain the blank symbol (meaning it wasn't visited yet by the head). Thus, we have computed $a_{i+1}(x)$, so we have all the information needed to compute the new embedding $o_{i+1}$.

To lower bound the all-layer margin of the constructed transformer, we use Theorem D.6, which requires existence of a "correction function" which can correct outputs in previous layers. Since we construct a network with intermediate layer entries in $\{0,1\}$, we can use the same correction function as Section 3.1, which rounds to the nearest bit. The full proof is provided in Section C.

## 5 Conclusion

This work proposes a new definition of approximation, statistically meaningful approximation, which ensures that the approximating family not only has sufficient expressivity, but also exhibits good statistical learnability. Towards a first analysis with this definition, we show approximability of two function classes: boolean circuits and Turing machines, with strong sample complexity guarantees depending only on the intrinsic properties of these function classes. There are several interesting directions to extend our study of statistically meaningful approximation. Examples include proving more upper and lower bounds for statistically meaningful approximation for different target functions and neural net architectures, and using our definition as a lens to compare architectures.

## 6 ETHICS AND REPRODUCIBILITY STATEMENTS

An ethics statement is not applicable for this work – this work is mainly theoretical and is several layers removed from empirical applications.

Section A contains the proofs for Section 2. Section B contains the formal construction and proof for boolean circuits. Section C contains the formal construction and proof for Turing machines. Section D rigorously introduces the correction function machinery and lower bounds the all-layer margin in terms of properties of the correction functions.

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

## A    PROOFS FOR SECTION 2

We prove Proposition 2.2 and Lemma 2.3.

*Proof of Proposition 2.2.* Let $(x_i)_{i=1}^n$ denote a $n$ i.i.d. training examples drawn from $P$ and fix $G \in \mathcal{G}$. Define $L(F) \triangleq \mathbb{E}_{x \sim P}[\bar{\ell}(F, x, G(x))]$ and $\widehat{L}(F) \triangleq \frac{1}{n} \sum_{i=1}^n \bar{\ell}(F, x_i, G(x_i))$. Let $\widehat{F} \in \mathcal{F}$ denote $\arg\min_{F \in \mathcal{F}} \widehat{L}(F)$, the empirical risk minimizer of $\widehat{L}$, which we aim to show has population loss for fitting $G$ bounded by $O(\epsilon + \frac{1}{\sqrt{n}})$. By standard arguments using Rademacher complexity, we have with probability $1 - \delta$,

$$\sup_{F \in \mathcal{F}} |L(F) - \widehat{L}(F)| \leqslant 2\mathrm{Rad}_{n,P}(\mathcal{L}_G) + \sqrt{\frac{\log(2/\delta)}{n}}$$

$$\leqslant 2\epsilon + \sqrt{\frac{\log(2/\delta)}{n}} \tag{A.1}$$

Now note that by the condition 3) on $\bar{\ell}$, there exists $F^\star$ with $L(F^\star) \leqslant \epsilon$. Now we have

$$L(\widehat{F}) - L(F^\star) \leqslant (L(\widehat{F}) - \widehat{L}(\widehat{F})) + (\widehat{L}(\widehat{F}) - \widehat{L}(F^\star)) + (\widehat{L}(F^\star) - L(F^\star))$$

We bound the first and last term in parenthesis by applying (A.1), and the middle term is bounded by 0, by definition of $\widehat{F}$. It follows that

$$L(\widehat{F}) - L(F^\star) \leqslant 4\epsilon + 2\sqrt{\frac{\log(2/\delta)}{n}}$$

$$\implies L(\widehat{F}) \leqslant 5\epsilon + 2\sqrt{\frac{\log(2/\delta)}{n}}$$

where we used $L(F^\star) \leqslant \epsilon$. Finally, we use the fact that $\bar{\ell}$ upper bounds $\ell$, so $\mathbb{E}_{x \sim P}[\ell(\widehat{F}(x), G(x))] \leqslant L(\widehat{F})$. Plugging in $\delta = 0.01$ gives the desired result. $\qquad\square$

*Proof of Lemma 2.3.* We first observe that $\bar{\ell}_\gamma(\theta, x, y) \leqslant \mathbb{1}(\rho(\theta, x, y) < \gamma)$ by definition, so by (2.3), for all $G \in \mathcal{G}$ we have

$$\inf_{\theta \in \Theta} \mathbb{E}_{x \sim P}[\bar{\ell}(\theta, x, G(x))] \leqslant \epsilon$$

Thus, it remains to check the Rademacher complexity condition for applying Proposition 2.2. Fixing any $G \in \mathcal{G}$, define the function class $\mathcal{L}_G$ as in Definition 2.1.

We first observe that following the same argument as Claim A.4 of (Wei & Ma, 2019b) (except we apply the perturbations to the parameters, rather than the hidden layers), $|\rho(\theta, x, y) - \rho(\theta', x, y)| \leqslant \|\theta - \theta'\|_2$ for any $\theta, \theta' \in \mathbb{R}^p$. Let $\mathcal{N}_{\|\cdot\|_2}(\varepsilon, \Theta)$ denote the $\varepsilon$-covering number of $\Theta$ in $\|\cdot\|_2$-norm, and $\mathcal{N}_{\|\cdot\|_\infty}(\varepsilon, \mathcal{L}_G)$ the $\varepsilon$-covering number of $\mathcal{L}_G$ in the norm defined by $\|H - H'\|_\infty = \max_{x \in \mathcal{X}} |H(x) - H'(x)|$ for any $H, H' \in \mathcal{L}_G$. The arguments of (Wei & Ma, 2019b) imply that $\log \mathcal{N}_{\|\cdot\|_\infty}(\varepsilon, \mathcal{L}_G) \leqslant \log \mathcal{N}_{\|\cdot\|_2}(\gamma\varepsilon, \Theta) \leqslant O\left(\left\lfloor \frac{\alpha^2 \log(p)}{\gamma^2 \varepsilon^2} \right\rfloor\right)$, where the last inequality is from standard covering number bounds for $\|\cdot\|_1$ balls. Now we can apply this covering number bound in the Dudley entropy integral, another standard step to bound Rademacher complexity, to obtain that for all $n$, $\mathrm{Rad}_{n,P}(\mathcal{L}_G) \lesssim \frac{\alpha \log n \sqrt{\log(p)}}{\gamma \sqrt{n}}$ (see arguments in (Wei & Ma, 2019b) for more detail). Solving for $n$ such that the r.h.s. of this equation is bounded by $\epsilon$ gives the desired result. $\quad\square$

Note that from the proof of Lemma 2.3, we would also obtain the following parameter-space all-layer margin generalization bound as a corollary, which may be of independent interest:

**Corollary A.1.** *In the setting of Lemma 2.3, let $Q$ denote a distribution over $(x, y)$ pairs, with $(x_i, y_i)_{i=1}^n$ denoting a set of $n$ i.i.d. training samples from $Q$. With probability $1 - \delta$ over the draw of the training samples, all classifiers $F(\cdot, \theta) \in \mathcal{F}$ which achieve zero 0-1 training loss satisfy*

$$\mathbb{E}_{x \sim Q}[\ell_{0\text{-}1}(F(x, \theta), y)] \leqslant O\left(\frac{\alpha\sqrt{\log(p)}}{\sqrt{n}} \sqrt{\frac{1}{n} \sum_{i=1}^n \frac{1}{\rho(\theta, x_i, y_i)^2}}\right) + \xi \tag{A.2}$$

*where $\xi \lesssim O\left(\frac{\log(1/\delta) + \log(n)}{\sqrt{n}}\right)$ is a low-order term.*

The proof of Corollary A.1 simply follows by plugging in the coverning number bound on $\rho$ derived in the proof of Lemma 2.3 into Lemma 2.2 of (Wei & Ma, 2019b).

# B    PROOFS FOR SECTION 3

This section completes the proof of Section 3. The following lemma formally states that we can construct the neural net to simulate the circuit layerwise.

**Lemma B.1.** *In the setting of Theorem 3.1, let $G$ denote the layered boolean circuit, which we aim to compute using a neural net. Let $G_i : \{0,1\}^{r_{i-1}} \to \{0,1\}^{r_i}$ denote function computed between the $i-1$-th and $i$-th layers of $G$, which we assume have $r_{i-1}$ and $r_i$ gates, respectively. Let $f$ denote the following 2-layer neural net architecture, parameterized by $\theta = (W_1, b_1, W_2, b_2)$:*

$$f(h,\theta) = \phi(W_2\phi(W_1 h + b_1) + b_2)$$

*Then there exist $\theta$ with $\|\theta\|_1 = O(r_i)$ such that for any $h \in \{0,1\}^{r_{i-1}}$,*

$$f(\widetilde{h}, \theta) = \widetilde{G_i(h)}$$

*where $\widetilde{h}$ takes $h$ and appends $w - r_{i-1}$ zeros, and likewise for $\widetilde{G_i}(h)$.*

We note that the proof of Lemma 3.2 follows by applying Lemma B.1 $q-1$ times. Using Lemma B.1, we can complete the proof of Theorem 3.1.

*Proof of Theorem 3.1.* Our proof will construct a neural network to compute any boolean circuit with all-layer margin lower bound $\frac{1}{\text{poly}(r,q)}$. By Lemma 2.3, this will be sufficient to guarantee meaningful approximation.

There are two steps in our construction: first, given any layered circuit $G \in \mathcal{G}_{q,r,s}$, we construct a neural net that directly simulates $G$ by computing the layers of $G$ one-by-one. Our construction shows that we can compute every layer in $G$ using two feedforward ReLU layers, and results in a neural net $\widehat{F}$ computing $G$, but with possibly small all-layer margin. The next step is to convert $\widehat{F}$ into a neural net with large all-layer margin, i.e., implement Lemma 3.3. To do this, we insert "correction functions" (Definition D.1) between every group of layers in $\widehat{F}$. These correction layers leverage the knowledge that unperturbed outputs of these layers should be contained in $\{0,1\}^w$ and perform elementwise rounding to map perturbed values back to $\{0,1\}^w$. Theorem D.6 formally shows that by introducing these correction layers can guarantee a lower bound on the all-layer margin roughly depending on the Lipschitz constants of each individual layer. Furthermore, each correction layer can be computed via two feedforward ReLU layers, so introducing the correction layers only increases depth by a constant factor.

We implement the proof plan by first applying Lemma B.1 $q$ times in order to obtain the function $\widehat{F}$ computing $G$ (with padding) mentioned above. The total $\|\cdot\|_1$-norm of the parameters so far is at most $s$. Now we use the correction function described in Proposition 3.4, which we apply coordinate-wise on non-padding coordinates. We apply the correction functions after each layer constructed in Lemma B.1. Note that each correction function requires at most double the width of the corresponding layer in the circuit, and the parameters for all correction functions add total $\|\cdot\|_1$-norm at most $O(s)$.

Note that at this point, minor modifications are still required in order to apply Theorem D.6. The neural net output is in $\{0,1\}^w$, not $\{-1,1\}$; we can remedy this by setting the last layer to compute the linear transformation $z \mapsto 2z - 1$ on the single non-padding coordinate corresponding to the output. Second, to make the depth of the architecture consistently $d$, we can add sequences of identity functions before this last linear layer just constructed, followed by correction layers, until each of the constructed approximating functions reaches the desired fixed depth $d$. This finally gives us parameters $\theta$ with $\|\cdot\|_1$-norm bound $O(s + d)$, so that the set of constructed functions is contained in $\mathcal{F}_{w,d,\alpha}$. Thus, we showed that for $G \in \mathcal{G}_{q,r,s}$, there exists $\theta$ such that $F(x,\theta) = 2G(x) - 1$ for all $x \in \{0,1\}^m$.

Finally, it is straightforward to check that Condition D.3 for Theorem D.6 is satisfied for Lipschitzness parameters which are polynomial in the circuit width $r$. Thus, we apply Theorem D.6 to obtain a lower bound $\widehat{\gamma} = \frac{1}{\text{poly}(r,q)} \geqslant \frac{1}{\text{poly}(s)}$ on the all-layer margin for every input $x \in \{0,1\}^m$. Finally, we directly apply Lemma 2.3 using $\gamma = \widehat{\gamma}$ to obtain the desired result.    $\square$

The following proposition will be used to construct basic gates in the circuit with a simple feedforward ReLU network.

**Proposition B.2.** *Let $x = \begin{bmatrix} x_1 \\ x_2 \end{bmatrix} \in \{0,1\}^2$ be binary inputs to* AND *and* OR *gates. The following feedforward ReLU networks compute the* AND *and* OR *functions: $F_{\text{AND}}(x) = \phi(x_1 + x_2 - 1)$, and $F_{\text{OR}}(x) = 1 - \phi(1 - x_1 - x_2)$.*

*Proof of Lemma B.1.* Each row of $W_1$ and value in $b_1$ will correspond to a single entry in the output of $\widetilde{G}_i$. The same applies for $W_2, b_2$. $W_2$ will be set to a diagonal matrix with entries in $\{-1, 0, 1\}$. For the 0 entries which only serve to pad the dimension, we set corresponding values in $W_1, b_1, W_2, b_2$ to be 0. For the remainder of the entries of $\widetilde{G}_i$ corresponding to actual gates in the circuit, in the case that the gates compute AND or OR, we fill in the values of corresponding rows in $W_1, b_1, W_2, b_2$ to implement the constructions for AND and OR in Proposition B.2. The construction for ID and NOT are even simpler. For example, to implement NOT$(z) = 1 - z$ for $z \in \{0, 1\}$ on coordinate $j$, we can set the $j$-th row of $W_1$ to have -1 on the diagonal and 0 everywhere else, $(b_1)_j = 1$, $(b_2)_j = 0$, and $(W_2)_{j,j} = 1$. It is easy to check that $\|\theta\|_1 = O(r_i)$ with this construction. $\qquad\square$

## C  PROOF OF THEOREM 4.1

### C.1  ADDITIONAL SETUP AND NOTATION

We fix any Turing machine $G \in \mathcal{G}$ and construct a transformer which can simulate $G$. Throughout this section, a superscript will be used to index layer indices, and a subscript to index timesteps.

We assume that the initial state of the tape has the input written at the left-most positions. The Turing machine always starts at a fixed initial state $z_{\text{init}}$. We let $[\varnothing] \in \mathcal{A}$ denote the blank symbol, which initially fills all positions on the tape which aren't part of the input. We construct a transformer that simulates the Turing machine up until it reaches a terminal state in $\mathcal{Z}_{\text{term}}$, at which the transformer will loop in that state until it hits a computation time $T$.

We introduce some notation which will appear throughout the construction. Define $w_{\text{pos}} \triangleq \lceil \log_2 T \rceil$. We use $w_{\text{pos}}$ to denote the effective dimension of the position embedding, as only $w_{\text{pos}}$ coordinates will be non-zero. For $0 \leqslant i \leqslant T$, define $\text{Bin}(i) \in \mathbb{R}^{w_{\text{pos}}}$ to be the vector containing the binary encoding of $i$: $\text{Bin}(i)_j = 1$ if the binary representation of $i$ contains 1 in the $j$-th bit and 0 otherwise.

For simplicity, the proof will focus on the setting without overparameterization, where we choose the dimension $w = w_{\text{TM}} \triangleq |\mathcal{Z}| + 2|\mathcal{A}| + 3w_{\text{pos}} + w_{\text{scr}}$ for storing all the hidden representations of the model, where $w_{\text{scr}} = O(w_{\text{pos}} + |\mathcal{A}| + |\mathcal{Z}|)$. We can extend our analysis to allow for arbitrary over-parameterization using $w > w_{\text{TM}}$ by designating a certain subset of the coordinates to always equal 0, and performing calculations using only a subset of $w_{\text{TM}}$ coordinates. We group the $w_{\text{TM}}$ coordinates using the following symbols: st for encoding the state, $\text{sym}_1$, $\text{sym}_2$ for encoding symbols, $\text{pos}_1$ and $\text{pos}_2$, $\text{pos}_3$ for encoding position, and scr, which is used as scratch space. Thus, for $h \in \mathbb{R}^w$, we can index its coordinates via the groups as follows:

$$
h = \begin{bmatrix}
h^{\text{st}} & \in \mathbb{R}^{|\mathcal{Z}|} \\
h^{\text{sym}_1} & \in \mathbb{R}^{|\mathcal{A}|} \\
h^{\text{sym}_2} & \in \mathbb{R}^{|\mathcal{A}|} \\
h^{\text{pos}_1} & \in \mathbb{R}^{w_{\text{pos}}} \\
h^{\text{pos}_2} & \in \mathbb{R}^{w_{\text{pos}}} \\
h^{\text{pos}_3} & \in \mathbb{R}^{w_{\text{pos}}} \\
h^{\text{scr}} & \in \mathbb{R}^{w_{\text{scr}}}
\end{bmatrix}
$$

When the meaning is clear from context, we use the superscript to index coordinate groups as described.

The position embedding $\beta(i)$ is defined formally so that $\beta(i)^{\text{pos}_1} = \text{Bin}(i)$, and $\beta(i)$ is 0 in all other coordinates. The encoder embedding matrix $E$ is such that

$$
\begin{aligned}
\text{Enc}_i(x)^{\text{sym}_1} &= \mathbb{1}_{|\mathcal{A}|}(x) \\
\text{Enc}_i(x)^{\text{pos}_1} &= \text{Bin}(i)
\end{aligned}
\tag{C.1}
$$

where $\text{Enc}_i(x)$ has 0's at all other coordinates. embedding function $e : \mathcal{A} \to \mathbb{R}^d$ for the encoder is defined such that $e(x)^{\text{sym}_1} = \mathbb{1}_{|\mathcal{A}|}(x)$, the one-hot encoding for $x \in \mathcal{A}$, and 0 everywhere else. We use $o_1, ..., o_T$ to refer to the output embeddings of the decoder. Our construction maintains the invariant that the output embedding $o_i$ encodes $z_i(x), a_i(x), l_i(x)$ for each $i$. To achieve this, we maintain

$$
\begin{aligned}
o_i^{\text{st}} &= \mathbf{1}_{|\mathcal{Z}|}(z_i(x)) \\
o_i^{\text{sym}_1} &= \mathbf{1}_{|\mathcal{A}|}(a_i(x)) \\
o_i^{\text{pos}_2} &= \text{Bin}(l_i(x))
\end{aligned}
\tag{C.2}
$$

and $o_i$ has 0 at all other coordinates. Thus, the input $o_i + \beta(i+1)$ to the decoder at step $i+1$ is of the form

$$
\begin{aligned}
(o_i + \beta(i+1))^{\text{st}} &= \mathbf{1}_{|\mathcal{Z}|}(z_i(x)) \\
(o_i + \beta(i+1))^{\text{sym}_1} &= \mathbf{1}_{|\mathcal{A}|}(a_i(x)) \\
(o_i + \beta(i+1))^{\text{pos}_1} &= \text{Bin}(i) \\
(o_i + \beta(i+1))^{\text{pos}_2} &= \text{Bin}(l_i(x))
\end{aligned}
\tag{C.3}
$$

## C.2 COMPLETING THE PROOF

We implement the first step 1) in Section 4.2 using the following lemma. Note that the lemma uses two consecutive feedforward ReLU layers, but in our actual proof we will simulate this using two transformer layers where the attention parameters are all $\mathbf{0}$, and only the feedforward layers are instantiated.

**Lemma C.1.** *Let $\mathcal{O}$ denote the set of decoder inputs in the form (C.3) encoding $z_{i-1}(x)$, $a_{i-1}(x)$, $l_{i-1}(x)$ for some timestep $i$. For parameters $\theta = (W_1, b_1, W_2, b_2)$, consider the following function computing a sequence of two feedforward ReLU layers: $f(h, \theta) = \phi(W_2 \phi(W_1 h + b_1) + b_2)$. There exist parameters $\theta$ such that for decoder inputs $h \in \mathcal{O}$,*

$$
\begin{aligned}
f(h, \theta)^{\text{st}} &= \mathbf{1}_{|\mathcal{Z}|}(z_i(x)) \\
f(h, \theta)^{\text{sym}_2} &= \mathbf{1}_{|\mathcal{A}|}(u_i(x)) \\
f(h, \theta)^{\text{pos}_1} &= \text{Bin}(i) \\
f(h, \theta)^{\text{pos}_2} &= \text{Bin}(l_{i-1}(x))
\end{aligned}
\tag{C.4}
$$

*Furthermore, $f(h, \theta)^{\text{scr}}$ will contain a one-hot encoding for $q_i(x)$, and besides this, $f(h, \theta)$ is 0 at all other coordinates. The parameters satisfy $\|\theta\|_1 = O(|\mathcal{Z}||\mathcal{A}| + w_{\text{pos}})$.*

*Proof.* We follow the construction used in Lemma B.2 of (Pérez et al., 2019). The first layer computes a one-hot encoding of the state, symbol input pair. We choose $W_1 : \mathbb{R}^{w_{\text{TM}}} \rightarrow \mathbb{R}^{|\mathcal{Z}||\mathcal{A}| + w_{\text{TM}}}$ so that the first $|\mathcal{Z}||\mathcal{A}|$ rows are described by:

$$
\begin{aligned}
(W_1)^{\text{st}}_{(z,a),:} &= \mathbf{1}_{|\mathcal{Z}|}(z) \\
(W_1)^{\text{sym}_1}_{(z,a),:} &= \mathbf{1}_{|\mathcal{A}|}(a)
\end{aligned}
$$

and 0 everywhere else. The remaining rows of $w_{\text{TM}}$ rows of $W_1$ simply implement the identity mapping. We choose $b_1$ so that its first $|\mathcal{Z}||\mathcal{A}|$ entries are -1, and all other entries are 0. We observe that from this construction, for all $h \in \mathcal{O}$ where $h$ encodes $z_{i-1}(x), a_{i-1}(x)$,

$$
\phi(W_1 h + b_1) = \begin{bmatrix} \mathbf{1}_{|\mathcal{Z}||\mathcal{A}|}((z_{i-1}(x), a_{i-1}(x))) \\ h \end{bmatrix}
$$

This is because before the ReLU, the first $|\mathcal{Z}||\mathcal{A}|$ entries of $W_1 h$ will have 2 on the $(z_{i-1}(x), a_{i-1}(x))$-th entry and be bounded by 1 everywhere else, so adding $\alpha_1$ and applying the activation will zero out all but one entry.

Now it is simple to pick $W_2$ so that $f(h, \theta)$ is as desired because we can construct it to exactly encode the output of $S(z, a)$ for each of its first $(z, a)$ columns and copy over the other necessary entries of $h$ as needed by (C.4). $\square$

The next lemma demonstrates that we can use an additional sequence of feedforward ReLU layers to produce $\text{Bin}(l_i(x))$, given $\text{Bin}(l_{i-1}(x))$ and $q_i(x)$.

**Lemma C.2.** *In the setting of Theorem 4.1 and Lemma C.1 above, there is a function $f$ parameterized by $\theta$ composed of $O(w_{\text{pos}})$ feedforward ReLU layers such that for any $h$ computed by the function in Lemma C.1 in the form (C.4) at timestep $i$,*

$$
\begin{aligned}
f(h, \theta)^{\text{st}} &= \mathbf{1}_{|\mathcal{Z}|}(z_i(x)) \\
f(h, \theta)^{\text{sym}_2} &= \mathbf{1}_{|\mathcal{A}|}(u_i(x)) \\
f(h, \theta)^{\text{pos}_1} &= \text{Bin}(i) \\
f(h, \theta)^{\text{pos}_2} &= \text{Bin}(l_{i-1}(x)) \\
f(h, \theta)^{\text{pos}_3} &= \text{Bin}(l_i(x))
\end{aligned}
\tag{C.5}
$$

*At all other coordinates, $F(h, \theta)$ takes value 0. Furthermore, the parameters satisfy $\|\theta\|_1 = O(w_{\text{pos}}(|\mathcal{Z}| + |\mathcal{A}| + w_{\text{pos}}))$.*

*Proof.* As the construction of Lemma C.1 encoded $q_i(x)$, the movement direction of the head, we can use feedforward ReLU layers to implement binary addition to either add or subtract 1 from $l_{i-1}(x)$. Let $v_1, v_2$ denote the bits in the scratch dimensions indicating the head movement, where $v_1 = 1, v_2 = 0$ indicates left and $v_1 = 0, v_2 = 1$ indicates right. Then more specifically, we first use $O(w_{\text{pos}})$ feedforward ReLU layers to compute $l_{i-1}(x) - v_1$, and then $O(w_{\text{pos}})$ additional feedforward ReLU layers to compute $l_{i-1}(x) - v_1 + v_2$. Note that the output would always be $l_i(x)$ by the definition of $v_1, v_2$.

It remains to implement a module which computes $\text{Bin}(j - v_1)$ given $v_1, \text{Bin}(j)$, and $\text{Bin}(j + v_2)$ given $v_2, \text{Bin}(j)$ for any $j \in [T]$. We can express the binary addition by a depth-$O(w_{\text{pos}})$ binary circuit, which can in turn be expressed by a neural net with $O(w_{\text{pos}})$ layers where each weight matrix has $\|\cdot\|_1$-norm $(|\mathcal{Z}| + |\mathcal{A}| + w_{\text{pos}})$ (which is required to implement the identity mapping to copy forward the other dimensions of $h$ which aren't involved in the binary addition). This gives the desired total $\|\cdot\|_1$-norm bound. □

The next lemmas implement steps 3), 4), 5) in Section 4.2. For the following lemmas, it will be helpful to further index the scratch dimensions as follows: for a vector $h \in w_{\text{scr}}$,

$$h^{\text{scr}} = \begin{bmatrix} h^{\text{scr}_1} \in \mathbb{R}^{|\mathcal{A}|} \\ h^{\text{scr}_2} \in \mathbb{R}^{|\mathcal{A}|} \\ h^{\text{scr}_3} \in \mathbb{R}^{w_{\text{pos}}} \\ h^{\text{scr}_4} \in \mathbb{R}^3 \end{bmatrix}$$

**Lemma C.3.** *In the setting of Theorem 4.1 and Lemma C.2 above, fix any timestep $i$ and define $i' = \max\{1 \leqslant t \leqslant i : l_{t-1}(x) = l_i(x)\}$. If $j$ such that $l_{t-1}(x) = l_i(x)$ exists, we define $i' = 0$ otherwise. Consider any $H_i = (h_1, ..., h_i)$, where $h_t$ is computed by the layer in Lemma C.2 for timestep $t$, and in the form* (C.5). *There is a function $f$ parameterized by $\theta$ consisting of $O(w_{\text{pos}})$ total self-attention and linear layers such that for all such $H_i$, the following holds:*

$$
\begin{aligned}
f(h_i, H_i, \theta)^{\text{st}} &= \mathbf{1}_{|\mathcal{Z}|}(z_i(x)) \\
f(h_i, H_i, \theta)^{\text{sym}_2} &= \mathbf{1}_{|\mathcal{A}|}(u_i(x)) \\
f(h_i, H_i, \theta)^{\text{pos}_1} &= \text{Bin}(i) \\
f(h_i, H_i, \theta)^{\text{pos}_2} &= \text{Bin}(l_{i-1}(x)) \\
f(h_i, H_i, \theta)^{\text{pos}_3} &= \text{Bin}(l_i(x)) \\
f(h_i, H_i, \theta)^{\text{scr}_1} &= \begin{cases} \mathbf{1}_{|\mathcal{A}|}(u_{i'}(x)) & \text{if } i' > 0 \\ \mathbf{0} & \text{otherwise} \end{cases} \\
F(h_i, H_i, \theta)^{\text{scr}_4}_1 &= \mathbb{1}(i' > 0)
\end{aligned}
\tag{C.6}
$$

*At all other coordinates, $F(H, \theta)$ takes value 0. Furthermore, the parameters satisfy $\|\theta\|_1 = O(w_{\text{pos}}(|\mathcal{Z}| + |\mathcal{A}| + w_{\text{pos}}))$.*

The proof plan will roughly implement a binary search to find $i'$, leveraging the attention layers. The first step in the binary search is to verify whether $i' > 0$, described below.

**Claim C.4.** *In the setting of Lemma C.3, let $H_i = h_1, ..., h_i$ be the input representations for timesteps $1, ..., i$. Suppose that each $h_t$ for $1 \leqslant t \leqslant i$ satisfies the following:*

$$
\begin{aligned}
h_t^{\text{pos}_1} &= \text{Bin}(t) \\
h_t^{\text{pos}_2} &= \text{Bin}(l_{t-1}(x))
\end{aligned}
\tag{C.7}
$$

*Additionally, suppose that $h_i$ is of the form in* (C.5). *Then there is a function $f^{(0)}$ parameterized by $\theta$ such that*

$$
\begin{aligned}
f^{(0)}(h_i, H_i, \theta)^{\text{scr}_1} &= \mathbf{0} \\
f^{(0)}(h_i, H_i, \theta)^{\text{scr}_3} &= \mathbf{0} \\
f^{(0)}(h_i, H_i, \theta)^{\text{scr}_4}_1 &= \mathbb{1}(i' > 0)
\end{aligned}
\tag{C.8}
$$

*The function $f^{(0)}$ can be computed by a single decoder self-attention layer with $\|\theta\|_1 = O(w_{\text{pos}})$.*

Next, we implement the binary search itself, using $w_{\text{pos}}$ self-attention layers. Each step of the binary search reveals a single bit of $i'$, so the $j$-th attention layer will compute a representation storing the $j$ most

significant bits of $i'$. We let $\mathrm{Bin}_j(l) \in \{0,1\}^{w_{\mathrm{pos}}}$ to denote the binary encoding of the $j$ most significant bits of $l$: $(\mathrm{Bin}_j(l))_{j'} = (\mathrm{Bin}(l))_{j'}$ for $1 \leqslant j' \leqslant j$, and $(\mathrm{Bin}_j(l))_{j'} = 0$ for $j' > j$. We also set $\mathrm{Bin}_0(l) = \mathbf{0}$. We use the superscript $^{(j)}$ to indicate the $j$-th set of layers in the binary search. The following claim implements each step of the binary search rigorously.

**Claim C.5.** *In the setting above and of Lemma C.3, let $H_i^{(j)} = h_1^{(j)}, \dots, h_i^{(j)}$ be the representations computed after the $j$-th group of layers for timesteps $1$ through $i$, for $0 \leqslant j \leqslant w_{\mathrm{pos}} - 1$. Suppose that each $h_t^{(j)}$ for $1 \leqslant t \leqslant i$ satisfies the following:*

$$
\begin{aligned}
h_t^{(j),\mathrm{pos}_1} &= \mathrm{Bin}(t) \\
h_t^{(j),\mathrm{pos}_2} &= \mathrm{Bin}(l_{t-1}(x))
\end{aligned}
\tag{C.9}
$$

*In addition, suppose that $h_i^{(j)}$ satisfies:*

$$
\begin{aligned}
h_i^{(j),\mathrm{scr}_1} &= \mathbf{0} \\
h_i^{(j),\mathrm{scr}_3} &= \begin{cases} \mathrm{Bin}_j(i') & \text{if } i' > 0 \\ \mathbf{0} & \text{otherwise} \end{cases} \\
(h_i^{(j),\mathrm{scr}_4})_1 &= \mathbb{1}(i' > 0)
\end{aligned}
\tag{C.10}
$$

*with all other coordinates matching the quantities prescribed in (C.5). Then there is a function $f^{(j+1)}$ parameterized by $\theta$ such that*

$$
\begin{aligned}
f^{(j+1)}(h_i^{(j)}, H_i^{(j)}, \theta)^{\mathrm{scr}_1} &= \mathbf{0} \\
f^{(j+1)}(h_i^{(j)}, H_i^{(j)}, \theta)^{\mathrm{scr}_3} &= \begin{cases} \mathrm{Bin}_{j+1}(i') & \text{if } i' > 0 \\ \mathbf{0} & \text{otherwise} \end{cases} \\
f^{(j+1)}(h_i^{(j)}, H_i^{(j)}, \theta)_1^{\mathrm{scr}_4} &= \mathbb{1}(i' > 0)
\end{aligned}
\tag{C.11}
$$

*with all other coordinates matching those prescribed in (C.5). We note that $f^{(j+1)}$ consists of a single decoder self-attention layer followed by single feedforward ReLU layer, with $\|\theta\|_1 = O(|\mathcal{Z}| + |\mathcal{A}| + w_{\mathrm{pos}})$.*

At the end of the $w_{\mathrm{pos}}$-th application of the binary search, we would have found $\mathrm{Bin}(i')$ exactly. It remains to apply another attention layer which attends directly to timestep $i'$ and copies $u_{i'}(x)$.

**Claim C.6.** *In the setting above and of Lemma C.3, let $H_i = h_1, \dots, h_i$ be the representations computed after the $w_{\mathrm{pos}}$-th group of layers constructed in Claim C.5 for timesteps $1$ through $i$. Suppose that each $h_t$ for $1 \leqslant t \leqslant i$ satisfies the following:*

$$
\begin{aligned}
h_t^{\mathrm{sym}_2} &= \mathbb{1}_{|\mathcal{A}|}(u_t(x)) \\
h_t^{\mathrm{pos}_1} &= \mathrm{Bin}(t) \\
h_t^{\mathrm{pos}_2} &= \mathrm{Bin}(l_{t-1}(x))
\end{aligned}
\tag{C.12}
$$

*In addition, suppose that $h_i$ satisfies:*

$$
\begin{aligned}
h_i^{\mathrm{scr}_1} &= 0 \\
h_i^{\mathrm{scr}_3} &= \begin{cases} \mathrm{Bin}(i') & \text{if } i' > 0 \\ \mathbf{0} & \text{otherwise} \end{cases} \\
(h_i^{\mathrm{scr}_4})_1 &= \mathbb{1}(i' > 0)
\end{aligned}
\tag{C.13}
$$

*with all other coordinates matching the quantities prescribed in (C.5). Then there is a function $f^{(w_{\mathrm{pos}}+1)}$ parameterized by $\theta$ such that $f^{(w_{\mathrm{pos}}+1)}(h_i, H_i, \theta)$ computes the desired output in (C.6). Furthermore, $f^{(w_{\mathrm{pos}}+1)}$ consists of a single decoder self-attention layer followed by a single feedforward ReLU layer, and $\|\theta\|_1 = O(|\mathcal{Z}| + |\mathcal{A}| + w_{\mathrm{pos}})$.*

Putting these together, we complete the proof of Lemma C.3.

*Proof of Lemma C.3.* For the purposes of this proof, we index the layers by a superscript to avoid confusion with indexing timesteps. We set $f^{(0)}$ to be the function defined in Claim C.4. We note that

layers output by $f^{(0)}$ satisfy the condition of Claim C.5, so we can apply Claim C.5 inductively to obtain layers $f^{(1)},...,f^{(w_{\text{pos}})}$ where their applying their composition results in representations satisfying (C.12) and (C.13). Now we set $f^{(w_{\text{pos}}+1)}$ to be the function constructed in Claim C.5, which gives the desired output. Finally, we note that by summing the $\|\cdot\|_1$ bounds for the parameters constructed in each layer, we can finally obtain $\|\theta\|_1 = O(w_{\text{pos}}(|\mathcal{Z}|+|\mathcal{A}|+w_{\text{pos}}))$. $\qquad\square$

We fill in the proofs of Claims C.4, C.5, and C.6 below.

*Proof of Claim C.4.* To construct the decoder self-attention, the query function will be of the form $Q(h) = W_Q h + b_Q$ and $K(h) = W_K h + b_K$, where $W_Q, W_K \in \mathbb{R}^{(w_{\text{pos}}+1)\times w}$ and $b_Q, b_K \in \mathbb{R}^{w_{\text{pos}}+1}$. We choose the parameters such that the following equations hold:

$$Q(h)_{1:w_{\text{pos}}} = 2h^{\text{pos}_3} - 1$$
$$Q(h)_{w_{\text{pos}}+1} = 1$$

and

$$K(h)_{1:w_{\text{pos}}} = 2h^{\text{pos}_2} - 1$$
$$K(h)_{w_{\text{pos}}+1} = 0$$

The value function $V(h)$ is such that $V(h)_1^{\text{scr}_4} = 1$, and $V(h)_\ell = 0$ on all other coordinates $\ell$, which can be implemented by a linear transformer. Finally, we set the null key $K_0$ and value $V_0$ such that $(K_0)_{w_{\text{pos}}+1} = w_{\text{pos}} - 1$, with 0 everywhere else, and $V_0 = 0$. Letting $\theta_{\text{attn}}$ denote the attention parameters, the layer is of the form

$$f^{(0)}(h_i, H_i, \theta) = \text{Attn}(h_i, H_i, \theta)$$

To see that $f^{(0)}$ satisfies (C.8), observe that if $i' > 0$, $Q(h_i)^\top K(h_{i'}) = w_{\text{pos}}$ by (C.7) and construction of $Q, K$. On the other hand, $Q(h_i)^\top K_0 = w_{\text{pos}} - 1$. Thus, $\text{argmax}_t Q(h_i)^\top K(h_t) \in [i]$, which implies that $f^{(0)}(h_i, H_i, \theta)_1^{\text{scr}_4} = 1$ by the construction of $V$. In the other case where $i' = 0$, we note that $Q(h_i)^\top K(h_t) \leqslant w_{\text{pos}} - 2$ for all $1 \leqslant t \leqslant i$, so the null position is attended to. By construction of $V_0$, this implies $f^{(0)}(h_i, H_i, \theta)_1^{\text{scr}_4} = 0$. As $V, V_0$ are 0 on all other coordinates, it follows that (C.8) holds. It's also easy to observe that the $\|\theta\|_1$ is as desired. $\qquad\square$

*Proof of Claim C.5.* The first layer in $f^{(j+1)}$ computes decoder self-attention. The query function is of the form $Q(h) = W_Q h + b_Q$, and the key function is of the form $K(h) = W_K h + b_h$, where $W_Q, W_K \in \mathbb{R}^{(w_{\text{pos}}+j+2)\times w}$ and $b_Q, b_K \in \mathbb{R}^{(w_{\text{pos}}+j+2)}$. We choose the parameters so that the following equations hold:

$$Q(h)_{1:w_{\text{pos}}} = 2h^{\text{pos}_3} - 1$$
$$Q(h)_{w_{\text{pos}}+1:w_{\text{pos}}+j} = 2h_{1:j}^{\text{scr}_3} - 1$$
$$Q(h)_{w_{\text{pos}}+j+1} = 1$$
$$Q(h)_{w_{\text{pos}}+j+2} = 1$$

and

$$K(h)_{1:w_{\text{pos}}} = 2h^{\text{pos}_2} - 1$$
$$K(h)_{w_{\text{pos}}+1:w_{\text{pos}}+j+1} = 2h_{1:j+1}^{\text{pos}_1} - 1$$
$$K(h)_{w_{\text{pos}}+j+2} = 0$$

Both of these functions can be constructed via linear transformations of $h$, with $\|W_Q\|_1 + \|W_K\|_1 + \|b_Q\|_1 + \|b_K\|_1 = O(w_{\text{pos}})$. Now we construct the value function $V(h) = W_V h + b_V$ such that $V(h)_3^{\text{scr}_4} = 1$ and $V(h)_\ell = 0$ on all other coordinates, which is also easily implemented by a linear layer. For the attention, the last quantities to construct are the null key $K_0$ and value $V_0$. $K_0$ will satisfy $(K_0)_{w_{\text{pos}}+j+2} = w_{\text{pos}} + j$, with 0 everywhere else. $V_0$ will simply be 0 on all coordinates. Letting $\theta_{\text{attn}} = (W_Q, b_Q, W_K, b_K, W_V, b_V, K_0, V_0)$ denote the attention parameters, the first layer will now be in the form

$$f^{(j+1),1}(h_i^{(j)}, H_i^{(j)}, \theta_{\text{attn}}) = \text{Attn}(h_i^{(j)}, H_i^{(j)}, \theta_{\text{attn}})$$

where Attn uses the constructed key, value, and query functions. We claim that $f^{(j+1),1}(h_i^{(j)}, H_i^{(j)}, \theta_{\text{attn}})$ satisfies the following:

$$f^{(j+1),1}(h_i^{(j)}, H_i^{(j)}, \theta_{\text{attn}})_3^{\text{scr}_4} = \begin{cases} 1 \text{ if } i' > 0 \text{ and has } (j+1)\text{-th bit } 1 \\ 0 \text{ otherwise} \end{cases} \tag{C.14}$$

For all other coordinates $\ell$, $f^{(j+1),1}(h_i^{(j)}, H_i^{(j)}, \theta_{\text{attn}})_\ell = (h_i^{(j)})_\ell$. To see this, we first observe that $Q(h_i^{(j)})^\top K_0 = w_{\text{pos}} + j$. Next, we observe that $Q(h_i^{(j)})_{1:w_{\text{pos}}}$ produces the encoding of $l_i(x)$ using binary $\{-1, +1\}$ bits, and $K(h_t^{(j)})_{1:w_{\text{pos}}}$ produces the encoding of $l_{t-1}(x)$ using binary $\{-1, +1\}$ bits by (C.9). In addition, $Q(h_i^{(j)})_{w_{\text{pos}}+1:w_{\text{pos}}+j} = 2\text{Bin}_j(i') - 1$ if $i' > 0$ and all 0's otherwise, and $K(h_t^{(j)})_{w_{\text{pos}}+1:w_{\text{pos}}+j+1} = 2\text{Bin}_{j+1}(t) - 1$. Note that by our construction, the maximum possible value of $Q(h_i^{(j)})^\top K(h_t^{(j)})$ is $w_{\text{pos}} + j + 1$, and the next largest possible value is $w_{\text{pos}} + j - 1$. Now there are 3 cases:

**Case 1:** $i' = 0$. In this case, we note that $l_i(x)$ never matches $l_{t-1}(x)$ for $1 \leqslant t \leqslant i$. Thus, by construction of the first $w_{\text{pos}}$ coordinates of $Q$ and $K$, the largest possible value of $Q(h_i^{(j)})^\top K(h_t^{(j)})$ is $w_{\text{pos}} + j - 1$, so the attention will always only attend to the null position, so the layer adds $V_0 = \mathbf{0}$ to $h_i^{(j)}$, preserving its value. Note that $(h_i^{(j),\text{scr}_4})_3 = 0$ in this case, which matches the desired behavior.

**Case 2:** $i' > 0$, and has $(j+1)$-th bit 0. In this case, we note that for all $t > i'$, $Q(h_i^{(j)})^\top K(h_t^{(j)}) \leqslant w_{\text{pos}} + j - 1$, because by definition such $t$ must satisfy $l_{t-1}(x) \neq l_i(x)$, so the first $w_{\text{pos}}$ coordinates contribute at most $w_{\text{pos}} - 2$ to the dot product. On the other hand, if $t \leqslant i'$, $t$ must have $(j+1)$-th bit 0, so $K(h_t^{(j)})_{w_{\text{pos}}+j+1} = -1$. This doesn't match the $(w_{\text{pos}} + j + 1)$-th bit of the query, so $Q(h_i^{(j)})^\top K(h_t^{(j)}) \leqslant w_{\text{pos}} + j - 1$ again. Thus, in this case, the null position is attended to again. The same reasoning as Case 1 then applies.

**Case 3:** $i' > 0$ and has $(j+1)$-th bit 1. In this case, $\max_t Q(h_i^{(j)})^\top K(h_t^{(j)}) = w_{\text{pos}} + j + 1$: for example, $t = i'$ achieves this maximum by our construction. As a result, the null position is not attended to. All the values in the positions attended to satisfy $V(h_t^{(j)})_3^{\text{scr}_4} = 1$, which matches the $(j+1)$-th bit of $i'$. Thus, (C.14) holds.

Finally, to complete the proof we simply append an additional feedforward ReLU layer which copies the value $f^{(j+1),1}(h_i^{(j)}, H_i^{(j)}, \theta_{\text{attn}})_3^{\text{scr}_4}$ to the output bit corresponding to the position indexed by $\cdot_{j+1}^{\text{scr}_3}$. This layer will also set the output bit corresponding to $\cdot_3^{\text{scr}_4}$ to 0. Note that these operations can be implemented with a linear layer, and applying a ReLU activation after won't change the output, which is in $\{0,1\}^w$. By (C.10), the constructed function will thus satisfy (C.11). It's also easy to observe that $\|\theta\|_1$ is as desired. $\qquad\square$

*Proof of Claim C.6.* The attention layer uses key and query functions which each compute linear transformations from $\mathbb{R}^w$ to $\mathbb{R}^{2w_{\text{pos}}+1}$. The value function is also linear. We choose parameters such that

$$Q(h)_{1:w_{\text{pos}}} = 2h^{\text{pos}_3} - 1$$
$$Q(h)_{w_{\text{pos}}+1:2w_{\text{pos}}} = 2h^{\text{scr}_3} - 1$$
$$Q(h)_{2w_{\text{pos}}+1} = 1$$

and

$$K(h)_{1:w_{\text{pos}}} = 2h^{\text{pos}_2} - 1$$
$$K(h)_{w_{\text{pos}}+1:2w_{\text{pos}}} = 2h^{\text{pos}_1} - 1$$
$$K(h)_{2w_{\text{pos}}+1} = 0$$

and

$$V(h)^{\text{scr}_1} = h^{\text{sym}_2}$$

Furthermore, we choose null keys and positions such that $(K_0)_{2w_{\text{pos}}+1} = 2w_{\text{pos}} - 1$, and $V_0 = \mathbf{0}$. To follow the attention layer, we construct a linear layer which simply zeros out coordinates indexed by $\cdot^{\text{scr}_3}$ and preserves all other coordinates. Note that because all outputs are either 0 or 1, applying a ReLU activation won't change the result. To see that this construction computes (C.6), we observe that if $i' > 0$, $Q(h_i)^\top K(h_{i'}) = 2w_{\text{pos}}$. Otherwise, if $i' = 0$, $Q(h_i)^\top K(h_t) \leqslant 2w_{\text{pos}} - 2$ for all $1 \leqslant t \leqslant i$. On the other hand, it always hold that $Q(h_i)^\top K_0 = 2w_{\text{pos}} - 1$. Thus, if $i' > 0$, the attention attends exactly to $i'$, so the value function satisfies

$V(h_{i'}) = \mathbf{1}_{|\mathcal{A}|}(u_{i'}(x))$, which would produce the output in (C.6), as desired. On the other hand, if $i' = 0$, the attention attends to the null position, so the attention layer sets $f^{(w_{\text{pos}}+1)}(h_i, H_i, \theta)^{\text{scr}_1} = \mathbf{0}$. Thus, $f^{(w_{\text{pos}}+1)}$ also produces the desired output in this case. It's also easy to observe that the $\|\theta\|_1$ is as desired. $\qquad\square$

The next step is to complete step 4) in Section 4.2 using encoder-decoder attention. The following lemma provides this construction.

**Lemma C.7.** *In the setting of Theorem 4.1 and Lemma C.3, consider any timestep $i$ and let $h$ denote an output of the function constructed in Lemma C.3, in the form* (C.6). *Let $e_1,...,e_m$ denote the outputs of the encoder, in the form* (C.1). *There is a function $f$ with parameter $\theta$ consisting of a single encoder-decoder attention layer such that for all such $h$ in the form* (C.6), *the following holds:*

$$
\begin{aligned}
f(h,(e_1,...,e_m),\theta)^{\text{st}} &= \mathbf{1}_{|\mathcal{Z}|}(z_i(x)) \\
f(h,(e_1,...,e_m),\theta)^{\text{sym}_2} &= \mathbf{1}_{|\mathcal{A}|}(u_i(x)) \\
f(h,(e_1,...,e_m),\theta)^{\text{pos}_1} &= \text{Bin}(i) \\
f(h,(e_1,...,e_m),\theta)^{\text{pos}_2} &= \text{Bin}(l_{i-1}(x)) \\
f(h,(e_1,...,e_m),\theta)^{\text{pos}_3} &= \text{Bin}(l_i(x)) \\
f(h,(e_1,...,e_m),\theta)^{\text{scr}_1} &= \begin{cases} \mathbf{1}_{|\mathcal{A}|}(u_{i'}(x)) & \text{if } i' > 0 \\ 0 & \text{otherwise} \end{cases} \\
f(h,(e_1,...,e_m),\theta)^{\text{scr}_2} &= \begin{cases} \mathbf{1}_{|\mathcal{A}|}(x_{l_i(x)}) & \text{if } l_i(x) \leqslant m \\ \mathbf{0} & \text{otherwise} \end{cases} \\
f(h,(e_1,...,e_m),\theta)^{\text{scr}_4}_1 &= \mathbb{1}(i' > 0) \\
f(h,(e_1,...,e_m),\theta)^{\text{scr}_4}_2 &= \mathbb{1}(l_i(x) \leqslant m)
\end{aligned}
\tag{C.15}
$$

*At all other coordinates, $f(h, (e_1, ..., e_m), \theta)$ takes value $0$. Furthermore, the parameters satisfy $\|\theta\|_1 = O(|\mathcal{A}| + w_{\text{pos}})$.*

*Proof.* We choose the encoder-decoder attention layer so that the key, value, and query functions are linear transformations. The key and query functions map $\mathbb{R}^w$ to $\mathbb{R}^{w_{\text{pos}}+1}$ and compute the following:

$$
\begin{aligned}
Q(h)_{1:w_{\text{pos}}} &= 2h^{\text{pos}_3} - 1 \\
Q(h)_{w_{\text{pos}}+1} &= 1
\end{aligned}
$$

and

$$
\begin{aligned}
K(h)_{1:w_{\text{pos}}} &= 2h^{\text{pos}_1} - 1 \\
K(h)_{w_{\text{pos}}+1} &= 0
\end{aligned}
$$

The value function computes

$$
\begin{aligned}
V(h)^{\text{scr}_2} &= h^{\text{sym}_1} \\
V(h)^{\text{scr}_4}_2 &= 1
\end{aligned}
$$

with 0's in all other coordinates. The null key $K_0$ satisfies $(K_0)_{w_{\text{pos}}+1} = w_{\text{pos}} - 1$, with 0's in all other coordinates. The null value $V_0$ satisfies $V_0 = \mathbf{0}$. We set

$$
f(h,(e_1,...,e_m),\theta) = \text{Attn}(h,(e_1,...,e_m),\theta)
$$

where Attn is the decoder-encoder attention using the key, value, and query described above. Now we observe that from this construction, if $h$ is in the form provided in (C.6), then $Q(h)_{1:w_{\text{pos}}} = \text{Bin}(l_i(x))$. In addition, we have $K(e_j)_{1:w_{\text{pos}}} = e_j^{\text{pos}_1} = \text{Bin}(j)$ for $1 \leqslant j \leqslant m$. Thus, by construction of $V, K_0, V_0$, if $l_i(x) \leqslant m$, the attention attends to position $l_i(x)$ in the embedding. The value function for this position satisfies $V(e_{l_i(x)})^{\text{scr}_2} = e_{l_i(x)}^{\text{sym}_1} = \mathbb{1}_{|\mathcal{A}|}(x_{l_i(x)})$. Thus, in this case $F(h,\theta)$ computes the desired output in (C.15). On the other hand, if $l_i(x) > m$, then the attention will attend to the null position, as $Q(h)^\top K_0 = w_{\text{pos}} - 1$, and the largest possible score for all other positions is $w_{\text{pos}} - 2$. In this case, (C.15) holds again. It is also easy to check that the desired bound on $\|\theta\|_1$ would hold. $\qquad\square$

Finally, we implement step 5) of the outline in Section 4.2 in the following lemma.

**Lemma C.8.** *In the setting of Theorem 4.1 and Lemma C.7, consider any timestep $i$ and any $h$ output by the function in Lemma C.7 taking the form in* (C.15). *Then there is a function $f$ with parameters $\theta$ consisting of a constant number of feedforward ReLU layers satisfying the following:*

$$f(h,\theta)^{\mathrm{st}} = \mathbf{1}_{|\mathcal{Z}|}(z_i(x))$$
$$f(h,\theta)^{\mathrm{sym}_1} = \mathbf{1}_{|\mathcal{A}|}(a_i(x)) \tag{C.16}$$
$$f(h,\theta)^{\mathrm{pos}_2} = \mathrm{Bin}(l_i(x))$$

*At all other coordinates, $F(h, \theta)$ takes values 0. Furthermore, the parameters satisfy $\|\theta\|_1 = O(|\mathcal{Z}| + |\mathcal{A}| + w_{\mathrm{pos}})$.*

*Proof.* It suffices to construct a sequence of layers which performs the following operations:

1) Compute the following vector $v \in \mathbb{R}^3$:

$$v = \begin{cases} \begin{bmatrix} 1 \\ 0 \\ 0 \end{bmatrix} & \text{if } h_1^{\mathrm{scr}_4} = 1 \\ \begin{bmatrix} 0 \\ h_2^{\mathrm{scr}_4} \\ 1 - h_2^{\mathrm{scr}_4} \end{bmatrix} & \text{if } h_1^{\mathrm{scr}_4} = 0 \end{cases}$$

Note that $v$ encodes the location of the symbol $a_i(x)$, as $a_i(x) = u_{i'}(x)$ if $i' > 0$, $a_i(x) = x_{l_i(x)}$ if $i' = 0$ and $l_i(x) \leq m$, and $a_i(x) = [\varnothing]$ otherwise. The vector $v$ is a one-hot vector indicating which of these three cases holds.

2) We can take $v_1$ and compute AND with all bits of $h^{\mathrm{scr}_1}$, which computes $\mathbf{1}_{|\mathcal{A}|}(u_{i'}(x)) = \mathbf{1}_{|\mathcal{A}|}(a_i(x))$ if $i' > 0$, and $\mathbf{0}$ otherwise.

3) We take $v_2$ and compute AND with all bits of $h^{\mathrm{scr}_2}$, which computes $\mathbf{1}_{|\mathcal{A}|}(x_{l_i(x)})$ if $v_2 = 1$, and $\mathbf{0}$ otherwise.

4) We take $v_3$ and compute AND with all bits of $\mathbf{1}_{|\mathcal{A}|}([\varnothing])$, which computes $\mathbf{1}_{|\mathcal{A}|}(a_i(x))$ if $v_3 = 1$.

5) We add the outputs of 2), 3), and 4) together, which gives $\mathbf{1}_{|\mathcal{A}|}(a_i(x))$. We copy this quantity into the output coordinates indexed by $\cdot^{\mathrm{sym}_1}$. Then we set coordinates not listed in (C.16) to 0, producing the desired output.

Each of these operations can be computed by a constant number of feedforward ReLU layers, with total parameter norm satisfying $\|\theta\|_1 = O(|\mathcal{Z}| + |\mathcal{A}| + w_{\mathrm{pos}})$. □

*Proof of Theorem 4.1.* We construct a neural net to compute any Turing machine with all-layer margin lower bound $\frac{1}{\mathrm{poly}(k,|\mathcal{A}|,\log T)}$ and apply Lemma 2.3 to turn this into a statement about statistically meaningful approximation.

For our Turing machine construction, we follow the outline laid out in Section 4.2. Fix any $G \in \mathcal{G}$. As mentioned, we first consider the case where $w = w_{\mathrm{TM}}$ exactly, as overparameterization is easy to deal with by always designating some subset of extra coordinates to be 0. We construct a transformer $\widehat{F}$ to compute $G$. First, we note that Lemma C.1 constructs a layer to compute the functionality described in 1). Next, the layer in Lemma C.2 performs the functionality in 2). Likewise, Lemmas C.3, C.7, C.8 construct layers which perform 3), 4), and 5). Thus, by applying the layers constructed from these lemmas in sequence, we obtain a transformer such that the output $o_T$ contains an onehot encoding for $z_T(x)$: $\mathbf{1}_{|\mathcal{Z}|}(z_T(x))$. We can now apply a linear weight vector $\theta_{\mathrm{cls}}$ on the output to obtain $\theta_{\mathrm{cls}}^\top o_T$, where $(\theta_{\mathrm{cls}})_z = 1$ for accept states $z \in \mathcal{Z}_{\mathrm{term}}$ and $(\theta_{\mathrm{cls}})_z = -1$ for reject states. For inputs $x \in \mathcal{X}$, by our construction this computes the desired $\mathrm{TM}(x)$. Next, following Theorem 3.1, we insert correction functions (Definition D.1) between every group of constructed layers, which can be implemented via two feedforward ReLU layers following Proposition 3.4. The parameters for all correction functions add total $\|\cdot\|_1$-norm at most $\mathrm{poly}(k,|\mathcal{A}|,\log T)$. Let $\widehat{F}(x,\widehat{\theta})$ denote the transformer constructed this way, with parameters $\widehat{\theta}$. Note that for all $x \in \mathcal{X}$, $\widehat{F}(x,\widehat{\theta}) = 2G(x) - 1$.

Next, there are several steps remaining to convert $\widehat{F}$ into the fixed architecture $F_{w,d,T}^{\text{tr}}$. First, we need to convert the layers in $\widehat{F}$ into transformer layers. This is achievable because every single decoder self-attention or encoder-decoder attention layer or feedforward ReLU module can be converted into a transformer layer by setting the two unused modules in the transformer layer to implement the identity function. This only increases the $\|\cdot\|_1$-norm by $\text{poly}(k,|\mathcal{A}|,\log T)$. Note that in particular, we can perform this conversion such that the correction functions form the last 2 feedforward ReLU layers in every transformer layer. The first 3 layers in the transformer layer correspond to ones constructed in the lemmas. Second, we need to expand the dimension to a consistent width $w$. This is achievable by padding each layer with coordinates designated to be 0, without affecting any of the $\|\cdot\|_1$-norm bounds on the parameters. Third, we need to expand the depth to a fixed depth $d$. We can achieve this by appending transformer layers which compute the identity function (and also include correction functions) as needed.

Now we aim to apply Theorem D.6 by viewing the transformer as a very deep network with depth $d = O(T\log T)$, by applying each of the steps in the transformer computation in sequence. Note that our construction for the transformer layers is such that we can view the self-attention, encoder-decoder attention, and single feedforward ReLU layer as a single function in the setting of Theorem D.6. The correction function corresponds to the last 2 feedforward ReLU layers in the transformer layer. (We observe that there are actually $m$ layers which depend on the input $x$, not a single layer $f_0$ as in the setting of Theorem D.6, but this is a minor difference where the same argument of Theorem D.6 still easily applies.) Note that this network uses layer-based weight sharing, which is handled by Theorem D.6. Furthermore, the depth of this network doesn't affect the all-layer margin because Theorem D.6 doesn't depend on the number of layers. We also observe that Condition D.4 holds for $\lambda = \text{poly}(|\mathcal{Z}|,|\mathcal{A}|,\log T)$, because all of the intermediate layers are sparse binary vectors with at most $|\mathcal{Z}|+|\mathcal{A}|+\log T$ nonzero entries.

Finally, it remains to check that Condition D.3 can hold for all of the defined layers for parameters that are polynomial in $|\mathcal{Z}|,|\mathcal{A}|,\log T$. This is straightforward to check for transformer layers where the attention layers have parameters $\mathbf{0}$, as standard results on the Lipschitzness of a single ReLU network would apply. For layers where the functionality comes from the attention mechanism, we observe that for valid inputs $x \in \mathcal{X}$, the largest attention score is always greater than the second largest by a margin of 1. Furthermore, ties only occur when all of the value vectors for the attended positions are already the same. As a result, the positions attended to by the layer will not change unless we perturb the parameters and inputs by $\Omega(\text{poly}^{-1}(|\mathcal{Z}|,|\mathcal{A}|,\log T))$. This reasoning can be used to conclude that Condition D.3 with Lipschitz constants $\text{poly}(|\mathcal{Z}|,|\mathcal{A}|,\log T)$, and distance parameters $\Omega(\text{poly}^{-1}(|\mathcal{Z}|,|\mathcal{A}|,\log T))$ holds. As a result, the all-layer margin bound from applying Theorem D.6 will also be $\Omega(\text{poly}^{-1}(|\mathcal{Z}|,|\mathcal{A}|,\log T))$, as desired. Finally, applying Lemma 2.3 with $\gamma = \Omega(\text{poly}^{-1}(|\mathcal{Z}|,|\mathcal{A}|,\log T))$ and using the fact that the parameter $\|\cdot\|_1$-norms are bounded by $\alpha$ gives the desired result. $\square$

## D  ALL-LAYER MARGIN LOWER BOUNDS VIA CORRECTION FUNCTIONS

We consider a generalized architecture for a $d$-layer network as follows. Let $f_0 : \mathcal{X} \times \Theta_0 \to \mathbb{R}^w$ map space of inputs $x \in \mathcal{X}$ and parameters $\theta \in \Theta_0$ to $w$-dimensional space. For simplicity we assume all intermediate layers have dimension $w$, and let $f_i : \mathbb{R}^w \times \Theta_i \to \mathbb{R}^w$ be the $i$-th function in the neural net for $d > i \geqslant 1$. We define $f_d$ to output values in $\mathbb{R}$. Let $\theta = (\theta_0,...,\theta_d) \in \Theta$ denote the full vector of parameters. The $i$-th hidden layer $h_i$ computes the following value, defined recursively:

$$h_0(x,\theta) = f_0(x,\theta_0)$$
$$h_i(x,\theta) = f_i(h_0(x,\theta),...,h_{i-1}(x,\theta),\theta_i)$$

The model computes output $h_d(x,\theta)$. We will assume the existence of "correction" functions $\zeta$ parameterized by $\xi = (\xi_0,...,\xi_{d-1}) \in \Xi_0 \times \cdot \times \Xi_{d-1}$ which correct errors in the model output for inputs $\mathcal{X}$:

**Definition D.1** (Correction functions). *Let $F' : \mathcal{X} \to \mathbb{R}$ be a model defined by layer functions $f_0,...,f_d$. Then $\zeta_0,...,\zeta_{d-1} : \mathbb{R}^w \to \mathbb{R}^w$, $\xi$ is a set of correction functions and parameters for $F'$, $\theta$ with radius $\sigma_\zeta$ if for all $i \in [d-1], x \in \mathcal{X}$ and $\widehat{h} \in \mathbb{R}^{\mathcal{X}}$ satisfying $\|\widehat{h} - h_i(x,\theta)\|_2 \leqslant \sigma_\zeta$,*

$$\zeta_i(\widehat{h},\xi_i) = h_i(x,\theta)$$

*We now define the function output $F$ with correction layers recursively by*

$$
\begin{aligned}
g_0(x,\theta,\xi) &= f_0(x,\theta_0) \\
\widetilde{h}_i(x,\theta,\xi) &= \zeta_i(g_{i-1}(x,\theta,\xi),\xi_i) \ \forall 0 \leqslant i \leqslant d-1 \\
g_i(x,\theta,\xi) &= f_i(\widetilde{h}_0(x,\theta,\xi),...,\widetilde{h}_{i-1}(x,\theta,\xi),\theta_i,\xi_i) \ \forall 1 \leqslant i \leqslant d \\
F(x,\theta,\xi) &= g_d(x,\theta,\xi)
\end{aligned}
$$

(D.1)

*We note that for all $x \in \mathcal{X}$, $F(x,\theta,\xi) = h_d(x,\theta)$.*

The key observation is that by adding correction layers to the model, we can transform a model with possibly small all-layer margin on the input data to one with large all-layer margin. We first need to characterize the Lipschitzness of the individual layers.

**Definition D.2.** *We say that a function $f(\cdot,\theta) : \mathcal{D} \rightarrow \mathcal{D}_{\text{out}}$ is $(\kappa_\theta, \mu, \sigma_h, \sigma_\theta)$-nice on $\mathcal{H} \subseteq \mathcal{D}$ with respect to $\|\cdot\|$ if the following hold:*

$$\|f(h,\theta) - f(h,\widehat{\theta})\|_2 \leqslant \kappa_\theta \|\theta - \widehat{\theta}\|_2 \max\{\|h\|, 1\} \qquad \forall \|\theta - \widehat{\theta}\| \leqslant \sigma_\theta, h \in \mathcal{H}$$

$$\|f(h,\widehat{\theta}) - f(\widehat{h},\widehat{\theta})\|_2 \leqslant \mu \|h - \widehat{h}\| \qquad \forall \|h - \widehat{h}\| \leqslant \sigma_h, \|\theta - \widehat{\theta}\| \leqslant \sigma_\theta, h \in \mathcal{H}$$

We will focus on the following norm on tuples of inputs $(v_1,...,v_i)$, where $h_j \in \mathbb{R}^w$ for all $j \in [i]$:

$$\|(v_1,...,v_i)\| = \max_j \|v_j\|_2 \tag{D.2}$$

We analyze the function $F$ output by a model with correction layers satisfying the following assumptions:

**Condition D.3.** *There are constants $\kappa_\theta, \kappa_\xi, \mu, \sigma_h, \sigma_\theta, \sigma_\zeta$ such that the following hold.*

*For $i \geqslant 1$, suppose that $f_i$ is $(\kappa_\theta, \mu, \sigma_h, \sigma_\theta)$-nice at $\theta_i$ on $(h_0,...,h_{i-1})(\mathcal{X})$ with respect to $\|\cdot\|$.*

*In addition, suppose that $f_0$ satisfies $\|f_0(x,\theta) - f_0(x,\widehat{\theta})\|_2 \leqslant \mu_0 \|\theta - \widehat{\theta}\|_2$ for all $x \in \mathcal{X}, \theta \in \Theta_0$.*

*Furthermore, suppose that for all $i$, $\zeta_i$ satisfies $\|\zeta_i(h,\xi_i) - \zeta_i(h,\widehat{\xi})\|_2 \leqslant \kappa_\xi \max\{\|h\|_2, 1\}\|\xi_i - \widehat{\xi}\|_2$ for all $\widehat{\xi}$ with $\|\xi_i - \widehat{\xi}\|_2 \leqslant \sigma_\xi$ and $h \in \mathbb{R}^w$.*

These conditions are all standard Lipschitzness-based conditions on the individual layer functions. Our lower bound for the all-layer margin will be expressed in terms of the constants here.

We will also need to assume a bound $\lambda$ on the norms of each of the layers computed by $h_i$.

**Condition D.4.** *The norms of the true layer values are bounded, that is, $\exists \lambda$ such that for all $0 \leqslant i \leqslant d$ and $x \in \mathcal{X}$,*

$$\max\{\|h_i(x,\theta)\|_2, 1\} \leqslant \lambda \tag{D.3}$$

We will also consider models with weight sharing, which allows our analysis to apply to architectures such as the transformer in Section 4.

**Definition D.5** (Layer-based weight sharing). *Let $\Theta' \subseteq \mathbb{R}^{w'}$, $\Theta_0 \subseteq \mathbb{R}^{w_0},...,\Theta_d \subseteq \mathbb{R}^{w_d}$ be some spaces of real-valued parameters. Suppose we wish to perform copying on parameters $\theta' \in \Theta'$ to produce parameters $\theta = (\theta_0,...\theta_d) \in \Theta = \Theta_0 \times \cdots \Theta_d$, where $\theta_i$ is the set of parameters given to layer function $f_i$. We say that a tuple of functions $\tau = (\tau_0,...,\tau_d) : \Theta' \rightarrow \Theta$ is a layer-based weight sharing scheme if each $\tau_i$ is of the form*

$$\tau_i(\theta') = (\theta'_{\pi_1},...,\theta'_{\pi_{b_i}}) \tag{D.4}$$

*where $\pi_1,...,\pi_{b_i}$ is a set of distinct indices taking values in $[w']$. Note that this ensures that parameters are not duplicated within a layer.*

We will now prove our main lower bound for the all-layer margin based on inserting correction functions at every layer.

**Theorem D.6.** *In the above setting, suppose that Conditions D.3 and D.4 hold for a function $F$ in the form given by* (D.1) *parametrized by $\theta$ with correction layers $\zeta_0,...\zeta_{d-1}$ parameterized by $\xi$ with correction radius $\sigma_\zeta < 1$. Suppose that $F(x) \in \{-1,+1\}$ $\forall x \in \mathcal{X}$. Then for all $x \in \mathcal{X}$, we can bound the all-layer margin of $F$ (defined in* (2.1)*) as follows:*

$$\rho_F((\theta,\xi),x,\mathbb{1}(F(x,\theta,\xi) \geqslant 0)) \geqslant \min\{\frac{\lambda}{\mu_0}, \frac{\sigma_\zeta}{\mu_0}, \sigma_\theta, \sigma_\xi, \frac{1}{2\kappa_\theta}, \frac{\sigma_\zeta}{2\kappa_\theta\lambda}, \frac{\sigma_h}{2\kappa_\xi\lambda}, \frac{\sigma_\zeta}{4\lambda\mu\kappa_\xi}, \frac{1}{4\mu\kappa_\xi}\} \tag{D.5}$$

*Here the subscript $F$ makes it explicit that the all-layer margin is for the architecture $F$. Furthermore, if we consider any layer-based weight-shared model $F'(x,\theta') \triangleq F(x,\tau^{(1)}(\theta'),\tau^{(2)}(\theta'))$ for valid weight-tying mappings $\tau^{(1)}$, $\tau^{(2)}$ (Definition D.5), the same bound holds for $\rho_{F'}(\theta',x,\mathbb{1}(F'(x,\theta') \geqslant 0))$.*

Our proof will first consider the case without weight sharing. We use $\widehat{\theta} = (\widehat{\theta}_0,...,\widehat{\theta}_d)$ and $\widehat{\xi} = (\widehat{\xi}_0,...,\widehat{\xi}_{d-1})$ to denote a perturbed set of parameter vectors. Furthermore, define the partially perturbed parameter sets $\widehat{\theta}_i \triangleq (\widehat{\theta}_0,...,\widehat{\theta}_i,\theta_{i+1},...,\theta_d)$ and $\widehat{\xi}_i \triangleq (\widehat{\xi}_0,...,\widehat{\xi}_i,\xi_{i+1},...,\xi_d)$. We also use $\widehat{\theta}_{-1} \triangleq \theta$ and $\widehat{\xi}_{-1} \triangleq \xi$ when convenient.

We consider perturbations such that the following norm bounds hold:

$$\|\widehat{\theta}_0 - \theta_0\|_2 \leqslant \min\{\frac{\lambda}{\mu_0}, \frac{\sigma_\zeta}{\mu_0}\} \tag{D.6}$$

$$\|\widehat{\theta}_i - \theta_i\|_2 \leqslant \min\{\sigma_\theta, \frac{1}{2\kappa_\theta}, \frac{\sigma_\zeta}{2\kappa_\theta\lambda}\} \tag{D.7}$$

$$\|\widehat{\xi}_i - \widehat{\xi}_i\|_2 \leqslant \min\{\sigma_\xi, \frac{\sigma_h}{2\kappa_\xi\lambda}, \frac{\sigma_\zeta}{4\lambda\mu\kappa_\xi}, \frac{1}{4\mu\kappa_\xi}\} \tag{D.8}$$

We show that such perturbations won't change the label predicted by the model, and so therefore the minimum of these quantities immediately gives a lower bound on the all-layer margin. Our proof will be by induction, with the following lemma providing the base case.

**Lemma D.7.** *In the setting of Theorem D.6, suppose that* (D.6) *holds. Then the following hold:*

$$\widetilde{h}_0(x,\widehat{\theta},\xi) = h_0(x,\theta)$$

$$\|g_0(x,\widehat{\theta},\widehat{\xi}) - h_0(x,\theta)\|_2 \leqslant \min\{\lambda,\sigma_\zeta\}$$

The next lemma provides the inductive step. Starting with the base case, we show that because of the presence of the correction functions, the perturbations with our given bounds won't change the next layer output by too much. This allows the correction function to fix the output of the next layer, and this argument can extend inductively.

**Lemma D.8.** *In the setting of Theorem D.6, fix some $1 \leqslant i \leqslant d$. Suppose that for all $0 \leqslant j < i$, it holds that for all $x \in \mathcal{X}$,*

$$\widetilde{h}_j(x,\widehat{\theta},\widehat{\xi}_{j-1}) = h_j(x,\theta) \tag{D.9}$$

*and*

$$\|g_j(x,\widehat{\theta},\widehat{\xi}) - h_j(x,\theta)\|_2 \leqslant \min\{\lambda,\sigma_\zeta\}$$

*In addition, suppose that $\widehat{\theta},\theta,\widehat{\xi},\xi$ satisfy* (D.7) *and* (D.8)*. Then it follows that for all $x \in \mathcal{X}$,*

$$\|g_i(x,\widehat{\theta},\widehat{\xi}) - h_i(x,\theta)\|_2 \leqslant \min\{\lambda,\sigma_\zeta\}$$

*Furthermore, for $1 \leqslant i \leqslant d-1$, we additionally have*

$$\widetilde{h}_i(x,\widehat{\theta},\widehat{\xi}_{i-1}) = h_i(x,\theta)$$

Combined, the two lemmas above allow us to inductively show that the prediction of the model is not changed whenever the perturbations are bounded by (D.6), (D.7), and (D.8). Next, we show that this translates directly to an all-layer margin lower bound.

**Lemma D.9.** *In the setting of Theorem D.6, suppose there exist norm bounds $a_0,...,a_d$, $b_0,...,b_{d-1}$ such that whenever $\|\widehat{\theta}_i - \theta_i\|_2 \leqslant a_i$ and $\|\widehat{\xi}_i - \xi_i\|_2 \leqslant b_i$, $|F(x,\theta,\xi) - F(x,\widehat{\theta},\widehat{\xi})| < 1$ for all $x \in \mathcal{X}$. Then we obtain the following lower bound on the all-layer margin, for all $x \in \mathcal{X}$:*

$$\rho_F((\theta,\xi),x,\mathbb{1}(F(x,\theta,\xi) \geqslant 0)) \geqslant \min\{a_0,...,a_d,b_0,...,b_{d-1}\}$$

*The same lower bound applies if we consider models that use layer-based weight sharing, defined by $F'(x,\theta') \triangleq F(x,\tau^{(1)}(\theta'),\tau^{(2)}(\theta'))$ for valid weight-tying mappings $\tau^{(1)}$, $\tau^{(2)}$ (Definition D.5).*

We can combine these steps to formally complete the proof of Theorem D.6.

*Proof of Theorem D.6.* Assuming the perturbation bounds (D.6) (D.7), and (D.8) hold, we can apply induction with Lemma D.7 as the base case and Lemma D.8 as the inductive step to conclude that $|F(x,\widehat{\theta},\widehat{\xi}) - F(x,\theta,\xi)| \leqslant \sigma_\zeta < 1$ for all $x \in \mathcal{X}$. We can now apply Lemma D.9 to obtain the desired bound on the all-layer margin. $\square$

We fill in the proofs of the supporting lemmas below.

*Proof of Lemma D.7.* By our definitions and Condition D.3, we have

$$\|g_0(x,\widehat{\theta},\widehat{\xi})-h_0(x,\theta)\|_2=\|f_0(x,\widehat{\theta}_0)-f_0(x,\theta_0)\|_2\leqslant\mu_0\|\theta_0-\widehat{\theta}_0\|_2\leqslant\min\{\lambda,\sigma_\zeta\}$$

Now we can apply the Definition D.1 of the correction function to get

$$\widetilde{h}_0(x,\widehat{\theta},\xi)=\zeta_0(g_0(x,\widehat{\theta},\widehat{\xi}),\xi_0)=h_0(x,\theta)$$

□

*Proof of Lemma D.8.* By expanding the expression for $h_i$, we observe that

$$h_i(x,\theta)=f_i(h_0(x,\theta),...,h_{i-1}(x,\theta),\theta_i)$$
$$=f_i(\widetilde{h}_0(x,\widehat{\theta},\xi),\widetilde{h}_1(x,\widehat{\theta},\widehat{\xi}_0)...,\widetilde{h}_{i-1}(x,\widehat{\theta},\widehat{\xi}_{i-2}),\theta_i) \tag{D.10}$$

We obtained the equality via (D.9). Now we write

$$g_i(x,\widehat{\theta},\widehat{\xi})=f_i(\widetilde{h}_0(x,\widehat{\theta},\widehat{\xi}),...,\widetilde{h}_{i-1}(x,\widehat{\theta},\widehat{\xi}),\widehat{\theta}_i) \tag{D.11}$$

We subtract the two expressions and add and subtract $f_i(\widetilde{h}_0(x,\widehat{\theta},\xi),\widetilde{h}_1(x,\widehat{\theta},\xi_0)...,\widetilde{h}_{i-1}(x,\widehat{\theta},\xi_{i-1}),\widehat{\theta}_i)$ to obtain

$$g_i(x,\widehat{\theta},\widehat{\xi})-h_i(x,\theta)=E_1+E_2$$

where

$$E_1\triangleq f_i(\widetilde{h}_0(x,\widehat{\theta},\widehat{\xi}),...,\widetilde{h}_{i-1}(x,\widehat{\theta},\widehat{\xi}),\widehat{\theta}_i)$$
$$-f_i(\widetilde{h}_0(x,\widehat{\theta},\xi),\widetilde{h}_1(x,\widehat{\theta},\widehat{\xi}_0)...,\widetilde{h}_{i-1}(x,\widehat{\theta},\widehat{\xi}_{i-2}),\widehat{\theta}_i)$$
$$E_2\triangleq f_i(\widetilde{h}_0(x,\widehat{\theta},\xi),\widetilde{h}_1(x,\widehat{\theta},\widehat{\xi}_0)...,\widetilde{h}_{i-1}(x,\widehat{\theta},\widehat{\xi}_{i-2}),\widehat{\theta}_i)$$
$$-f_i(\widetilde{h}_0(x,\widehat{\theta},\xi),\widetilde{h}_1(x,\widehat{\theta},\widehat{\xi}_0)...,\widetilde{h}_{i-1}(x,\widehat{\theta},\widehat{\xi}_{i-2}),\theta_i)$$

We first bound $E_1$. We note that for all $0\leqslant j\leqslant i-1$

$$\|\widetilde{h}_j(x,\widehat{\theta},\widehat{\xi})-\widetilde{h}_j(x,\widehat{\theta},\widehat{\xi}_{j-1})\|_2=\|\zeta_j(g_j(x,\widehat{\theta},\widehat{\xi}),\widehat{\xi}_j)-\zeta_j(g_j(x,\widehat{\theta},\widehat{\xi}),\xi_j)\|_2$$
$$\leqslant\kappa_\xi\max\{\|g_j(x,\widehat{\theta},\widehat{\xi})\|_2,1\}\|\widehat{\xi}_j-\xi_j\|_2$$

The last inequality used Condition D.3 and $\|\widehat{\xi}_j-\xi_j\|_2\leqslant\sigma_\xi$. Now defining $H'\triangleq(\widetilde{h}_0(x,\widehat{\theta},\widehat{\xi}),...,\widetilde{h}_{i-1}(x,\widehat{\theta},\widehat{\xi}))$ and $H\triangleq(\widetilde{h}_0(x,\widehat{\theta},\xi),\widetilde{h}_1(x,\widehat{\theta},\widehat{\xi}_0)...,\widetilde{h}_{i-1}(x,\widehat{\theta},\widehat{\xi}_{i-2}))$, it follows that

$$\||H-H'\||=\max_{0\leqslant j\leqslant i-1}\kappa_\xi\max\{\|g_j(x,\widehat{\theta},\widehat{\xi})\|_2,1\}\|\widehat{\xi}_j-\xi_j\|_2$$

Plugging in $\|g_j(x,\widehat{\theta},\widehat{\xi})\|_2\leqslant\|h_j(x,\theta)\|_2+\|g_j(x,\widehat{\theta},\widehat{\xi})-h_j(x,\theta)\|_2\leqslant2\lambda$, $\lambda\geqslant1$, and $\|\widehat{\xi}_j-\xi_j\|_2\leqslant\frac{\sigma_h}{2\kappa_\xi\lambda}$, we obtain $\||H-H'\||\leqslant\sigma_h$. Furthermore, we note that $H\in(h_0,...,h_{i-1})(\mathcal{X})$, so we can apply Condition D.3 and Definition D.2 to obtain

$$\|E_1\|_2=\|f_i(H',\widehat{\theta}_i)-f_i(H,\widehat{\theta}_i)\|_2$$
$$\leqslant\mu\||H-H'\|| \qquad\qquad (\text{since }\|\widehat{\theta}_i-\theta_i\|_2\leqslant\sigma_\theta\text{ and }\||H-H'\||\leqslant\sigma_h)$$
$$\leqslant2\lambda\mu\kappa_\xi\max_j\|\widehat{\xi}_j-\xi_j\|_2$$

Next, we bound $E_2$ by applying Condition D.3 and Definition D.2 again, using $\|\widehat{\theta}_i-\theta_i\|_2\leqslant\sigma_\theta$:

$$\|E_2\|_2=\|f_i(H,\widehat{\theta}_i)-f_i(H,\theta_i)\|_2$$
$$\leqslant\kappa_\theta\|\widehat{\theta}_i-\theta_i\|_2\max\{\||H\||,1\}$$
$$=\kappa_\theta\|\widehat{\theta}_i-\theta_i\|_2\max\{\|h_j(x,\theta)\|_2\}_j\cup\{1\}$$
$$\leqslant\kappa_\theta\|\widehat{\theta}_i-\theta_i\|_2\lambda$$

where we applied Condition D.4. By triangle inequality, follows that

$$\|g_i(x,\widehat{\theta},\widehat{\xi})-h_i(x,\theta)\|_2\leqslant\|E_1\|_2+\|E_2\|_2$$
$$\leqslant\kappa_\theta\|\widehat{\theta}_i-\theta_i\|_2\lambda+2\lambda\mu\kappa_\xi\max_j\|\widehat{\xi}_j-\xi_j\|_2$$

Now by the assumptions on $\|\widehat{\theta}_i - \theta_i\|_2$ and $\|\widehat{\xi}_j - \xi_j\|_2$, we can check that the r.h.s. is bounded by $\min\{\lambda, \sigma_\zeta\}$. Finally, we note that by Definition D.1 of the correction function, we have

$$\widetilde{h}_i(x, \widehat{\theta}, \widehat{\xi}_{i-1}) = \zeta_i(g_i(x, \widehat{\theta}, \widehat{\xi}), \xi_i) = h_i(x, \theta)$$

where we used the fact that $\|g_i(x, \widehat{\theta}, \widehat{\xi}) - h_i(x, \theta)\|_2 \leqslant \sigma_\zeta$. $\qquad\square$

*Proof of Lemma D.9.* Note that if $\|(\theta, \xi) - (\widehat{\theta}, \widehat{\xi})\|_2 < \bar{a} \triangleq \min\{a_0, ..., a_d, b_0, ..., b_{d-1}\}$, then by the conditions of the lemma, $|F(x, \theta, \xi) - F(x, \widehat{\theta}, \widehat{\xi})| < 1$. However, because $F(x, \theta, \xi) \in \{-1, +1\}$ for all $x \in \mathcal{X}$, the sign of the output is unchanged, which means $F(x, \theta, \xi) F(x, \widehat{\theta}, \widehat{\xi}) > 0$. This means that we must perturb $(\theta, \xi)$ by $\|\cdot\|_2$-norm at least $\bar{a}$ to satisfy the constraint in the all-layer margin definition, giving us the lower bound. We note that a similar argument applies to layer-based weight sharing because there are no parameters shared within a layer, so if the perturbation to $\theta'$ has $\ell_2$ norm less than $\bar{a}$, the parameters in $\tau^{(1)}(\theta')$, $\tau^{(2)}(\theta')$ will also have a perturbation of at most $\bar{a}$ in each layer. The same reasoning as before then applies. $\qquad\square$

