# OpenReview forum: "Statistically Meaningful Approximation: a Theoretical Analysis for Approximating Turing Machines with Transformers"
_ICLR.cc/2022/Conference — ICLR 2022 Submitted_

### Official Review · Reviewer_Lqbr · 2021-10-27

**Correctness:** 4
**Technical Novelty And Significance:** 3
**Empirical Novelty And Significance:** Not applicable
**Recommendation:** 6
**Confidence:** 4

**Main Review:**

Strength:

1. The concept of SM approximation unifies approximation and generalization, which is novel and maybe interesting.

2. The paper actually shows that when optimizing the all-layer margin loss, both fully connected nets and transformers would "prefer" to learn some simple function class using a relatively small number of samples. This is interesting because we may expect different surrogate losses for different network structures.


Below are my concerns:

- If we already know that $\mathcal{F}$ can exactly represent $\mathcal{G}$, then we can choose the surrogate loss $\bar{l}$ such that it outputs infinite loss when $F \notin \mathcal{G}$. In this sense, optimizing the surrogate loss would be just doing empirical risk minimization within $\mathcal{G}$. Using this somewhat cheating trick, showing SM approximation reduces to (1). show that $\mathcal{F}$ can exactly represent $\mathcal{G}$, and (2). provide a generalization bound for $\mathcal{G}$. Therefore, to some extent, the nice sample complexity results in the paper are not that surprising.

- minor point: It would be good if the surrogate loss (all-layer-margin loss as proposed in the paper) is actually computationally tractable. In other words, a good future direction would be to try to show GD/SGD intrinsically optimize this loss.

**Summary Of The Paper:**

The paper proposes a novel concept called statistically meaningful approximation. Roughly speaking, SM approximation means that performing empirical risk minimization over some properly chosen surrogate loss would result in a hypothesis with small population risk. Then the paper develops necessary tools and proves two results: (1). fully-connected neural networks can SM approximate boolean circuits with a sample size that is poly in the number of gates and $\log$ in the width and depth of the circuit; (2). transformers can SM approximate Turing machines with a sample size that is poly in the number of states, alphabet size, and the log of the computation steps.

**Summary Of The Review:**

The theory of the paper is novel and interesting.

---

### Official Review · Reviewer_RNAM · 2021-11-01

**Correctness:** 3
**Technical Novelty And Significance:** 2
**Empirical Novelty And Significance:** Not applicable
**Recommendation:** 5
**Confidence:** 2

**Main Review:**

The presentation is clear and the proofs seem adequately proven.
But the novelty and importance of the new definition is not adequately demonstrated for this reviewer.
Statistical learning theory is a vast field and the ingredients of the definition are not new.
The two examples are improvements on previous work which, it appears,
could have been applied with previous phrasings of generalization and approximation accuracy.
Also the definition is only constructive on the existence side (F SM-approximates G if "there exists a loss ...")
and not on the non-existence side.  It would be reassuring to see the proof of a negative example.
I think much more discussion is required as well as much more review of previous related work for this to become more than just another definition added to a long list.

**Summary Of The Paper:**

This paper proposes a definition of "statistically meaningful approximation" or SM-approximation which combines learnability, generalization ability and expressivity (approximation accuracy) in a straightforward way.  Two theorems on SM-approximation are formulated and proven, one for approximating Boolean circuits and one for approximating a general Turing machine computation with a transformer network.  It is argued that the new definition has technical advantages.

**Summary Of The Review:**

Interesting proposal but not given enough motivation or set in enough context to be convincing.

---

### Official Review · Reviewer_eLwz · 2021-11-03

**Correctness:** 3
**Technical Novelty And Significance:** 2
**Empirical Novelty And Significance:** 3
**Recommendation:** 5
**Confidence:** 4

**Main Review:**

1. Regarding the notion of SM approximation, it seems a ``re-branding'' of the classical generalization theory. Specifically, as shown in the Proof of Proposition 2.2, the trick is to (i) find a surrogate loss function $\bar \ell$ that dominates the loss function $\ell$ and (ii) apply the standard generalization theory to $ \bar \ell$. Here the surrogate loss function is a ramp loss combined with the all-layer margin. I would suggest modifying the claim and giving more credit to the classical generalization theory.

2. The theoretical results and the proofs seem not sufficiently rigorous. For example, in theorem 4.1, the statement requires $d \asymp \log T$, it seems unclear how large the hidden constant is. In the proof, there are a lot of $O(\cdot)$ and $\mathrm{poly}$ notations that might hide very large constants.

3. The authors treat the Turing machine as a binary classification problem and uses transformer layers to represent the Turing machine. However, such a classification perspective seems to neglect the fact that the Turing machine actually involves a sequence of input and outputs, and thus using a single binary output might be problematic.

4. The all-layer margin and the margin loss in (2.2) seem hard to compute. Also, it seems intractable to find the global minimizer of the empirical risk. Thus, it is not clear how this work is relevant to practical deep learning. It would be great to have empirical results that approximately compute the global minimizer of the empirical loss.

5. In terms of the contribution, it seems that the major contribution of this work is to represent the Turing machine using neural networks, compute its all-layer margin, and apply the standard generalization theory. Maybe it is better to downplay the first point in the summary of contributions in introduction:
> our contributions are: 1) we propose a new notion of statistically meaningful approximation,
intended to provide more meaningful approximation guarantees by requiring that the approximating family
have good statistical learnability;

6. Related works:

   a. Nonparametric regression using deep neural networks with ReLU activation function

   b. Sparse-Input Neural Networks for High-dimensional Nonparametric Regression and Classification

   c. Generalization Bounds of Stochastic Gradient Descent for Wide and Deep Neural Networks

   d. Generalization error bounds of gradient descent for learning over-parameterized deep ReLU networks


**Summary Of The Paper:**

This paper studies the approximation of boolean circuits and Turing machines using neural networks. To this end, the authors propose a new notion of approximation criteria -- statistically meaningful (SM) approximation -- that specifies that the function class is both learnable and has a small bias in the sense of classical approximation theory. For both the boolean circuits and Turing machines, the authors explicitly construct two classes of neural networks, based on feedforward network and transformers respectively, that achieve SM approximation.

**Summary Of The Review:**

This work seems to only establish a generalization theory for using neural networks to learn boolean functions and Turing machines. The main contribution is showing that proper neural network structures can represent the target functions and the networks have large margins However, the estimator (and even the surrogate loss function) are not easy to compute, which limits the practicality of this work. In addition, treating Turing machine as a classifier seems restrictive.

---

### Official Review · Reviewer_VpYu · 2021-11-04

**Correctness:** 3
**Technical Novelty And Significance:** 3
**Empirical Novelty And Significance:** Not applicable
**Recommendation:** 3
**Confidence:** 4

**Main Review:**

There are already many proofs that functions and programs can be represented by various classes of neural networks ("universality theorems", notably classical work by Cybenko/Barron which approximate arbitrary functions by depth 2 non-linear networks and recent work by Eldan-Shamir/Telgarsky showing the trade-offs between depth and width). This paper argues that some existing results rely in infinite precision and are thus undesirable. (It seems to me the latter objection only holds for specific universality theorems). The proposed criterion gets around this by saying that the representation should come with a loss function whose empirical minimizer captures the function of interest. This is simple, intuitive and perhaps obvious so far. Is it useful? I am not sure.

So, to demonstrate its value, they show that Boolean circuits can be represented "statistically meaningfully", i.e., they provide a loss function and a proof that an empirical minimum on a finite sample must give an honest representation. What this comes down to is showing that the representation parametrization can be done with a large margin, and the loss function focus on finding a large margin, so that empritical minima must find valid solutions. I was able to follow the construction for Boolean circuits (note, these are layered Boolean circuits, this should be mentioned up front) by relying on ReLUs and a simple trick to lift into higher-dimensional non-linear features and create a significant margin even if one doesn't exist in the first place. The particular loss function proposed is apparently a variant of a recent proposal (but was new to me).

I could not follow the second application (transformers can represent Turing machines). Unfortunately, while I understand the latter is a hot topic, at the least the authors should define the class of functions they are dealing with, explain their representation power and advantages before going to their statistically meaningful construction.

**Summary Of The Paper:**

This paper proposes a new criterion for learnable representations of Boolean circuits and Turing machines. The criterion "statistically meaningful" is the main contribution along with its application to the above two classes.

**Summary Of The Review:**

Presumably the value of this paper is conceptual; I don't see any benefits to ML algorithms (e.g., the loss function proposed is not robust, i.e., some errors in the data labels will cause it fail). While the definition is simple and natural, I am afraid I don't see the conceptual advantages. There are already existing, more powerful definitions that provide some insight into computation, e.g., Statistical-Query Learnable, where every query is answered empirically up to some tolerance.

---

### Decision · Program_Chairs · 2022-01-20

**Decision:**

Reject

**Comment:**

This is an extremely interesting and timely paper regarding the approximation ability, with statistical consequences, of circuits and (computation-bounded) Turing machines by feedforward networks and transformers. The paper has an interesting and valuable setting, and also many unusual ideas, together which can inspire a lot of future work.  Unfortunately, the reviewers had significant difficulties with the presentation and setting; the Transformer material in particular lacks clarity.  As such, the paper could use more time and polish.

Separately, I will recommend in the future that authors consider making use of the rebuttal and revision phase.  While it is not strictly required, it seems that in ICLR, scores shift quite a lot in those phase, and it has (for better or worse) become standard to have a thorough involvement in this phase.  It was difficult to cause score changes after the initial phase due to the lack of review responses.  That said, I sincerely hope the authors continue with this valuable line of work.